



# Climate and Disease: the connection between temperature values and precipitation rates and the probability of death due to waterborne and airborne diseases in historical urban space (the evidence from Poznań, Poland)

Grażyna Liczbińska[1], Jörg Peter Vögele[2], Marek Brabec[3,4]

[1]Institute of Human Biology and Evolution, Faculty of Biology, Adam Mickiewicz University, Poznań, 61–614 Poznań, Poland

[2] Department of History, Philosophy and Ethics of Medicine, University of Düsseldorf, Germany

[3]Department of Statistical Modelling, Institute of Computer Science, The Czech Academy of Sciences, Prague, Czech Republic

[4]Department of Biostatistics, National Institute of Public Health, Prague, Czech Republic

*Correspondence to*: Grażyna Liczbińska (grazyna@amu.edu.pl)

**Abstract.** The study examines the relationship between temperature values and precipitation rates as explanatory variables and the probability of death due to waterborne, airborne, and other diseases in historical urban space. So far, the literature has not been focused on epidemiology of 19th -century Polish urban areas in climatological context. We used individual data

on mortality from Poznań parish death registers for 1850–1900. Each deceased individual was assigned the average monthly temperature values and precipitation rates in the month of death, LAG1 temperature and LAG1 rainfall, and a quarter of residence. We studied the relationship between weather conditions and mortality using formalized statistical models reflecting the discrete nature of the response data (via multinomial logistic regression). Lagged monthly average temperature values and lagged monthly average precipitation rates were better predictors of airborne and waterborne disease mortality than

the concurrent (non-lagged) monthly averages. The lagged effects of temperature and precipitation on waterborne and airborne diseases were significant (except for the smooth lagged average monthly temperature effect for airborne diseases). There was also significant spatial heterogeneity (differences among city quarters) in the prevalence of deaths due to waterborne and airborne diseases.

## 1 Introduction

Humans inhabiting a given climate zone for generations have developed particular characteristic traits that make them better suited to the environment. Due to genetic changes, individuals show adaptations in the structure and/or functioning of their organisms allowing them to live successfully in various environments. Genetic adaptations, which give humans well-developed thermoregulation system by changes in body temperature within the narrow temperature intervals, and civilization



achievements (clothing, buildings, heating, air conditioning, etc.), have enabled them to cope with climatic stresses and
pressures.

Meanwhile, more and more researchers have been detecting the existence of a relationship between climate, or more precisely, its components such as temperature and precipitation, and the health status and well-being of humans. When examining the relationship between climate and health, attention is also paid to the consequences of these changes over time. Extreme weather events (heat and cold waves), floods, and droughts lead to changes in the ecology of infectious diseases
(Patz et al., 1996), and have a strong impact on changes in health and disease profiles across the world and inhibit the potential for declines in morbidity and mortality (Levy et al., 2018). Climate changes might lead to the emergence of new pathogens, the increased incidence of old pathogens, and/or the re-emergence of infectious diseases (Campbell-Lendrum et al., 2009; Cascio et al., 2011; Barra et al., 2014).

The relationship between temperature values and precipitation rates is observed in correlations between various
disease rates, cause-specific death rates, and weather variations on different time scales – over weeks, months, or years. Recently, many countries in temperate climate zones have dealt with the phenomenon of extremely high ambient temperatures. Prolonged periods of hot days are responsible for the increase in heat-related death rates (e.g., Hajat et al., 2007; D'lppoliti et al., 2010; Tong et al., 2012; Kent et al., 2014; Chen et al., 2015), especially due to cardiovascular and respiratory diseases (Basu and Samet, 2002; Donaldson et al., 2003; Basu, 2009), mental and nervous systems disorders (Stafoggia et al., 2006),
diabetes, and kidney and urinary system diseases (Conti et al., 2005; Rey et al., 2007). Recent studies have also suggested a link between extremely high ambient temperature during pregnancy and high prevalence of hypertension, eclampsia, and cataract in infants (Poursafaet al., 2015), as well as perinatal brain injury (Kasdorf and Perlman, 2013), and a high prevalence of preterm births (Lajinian et al., 1997; Flouris et al., 2009; Basu et al., 2010; Auger et al., 2014; Ha et al., 2017). A significant increase in infant mortality during extremely high ambient temperatures was observed in Great Britain (Rooney
et al., 1998), South Korea (Kysely and Kim, 2009), Spain (Basagaňa et al., 2011), and Australia (Nitsche et al., 2011).

Infectious agents such as viruses, bacteria, and protozoa, and their vector organisms (e.g., ticks and mosquitoes) are devoid of thermostatic mechanisms, and hence their reproduction and survival rates are strongly affected by fluctuations in temperature (Patz et al., 2005). Weather conditions, shaped by rainfall and high temperatures, have a fairly well-documented impact on the increase in the incidence of infectious diseases (e.g., Shiffman et al.,1976; Curriero et al., 2001; Cann et al.,
2013) and on the increase of their ranges of occurrence, as in the case of such diseases as malaria, dengue fever, dengue haemorrhagic fever, and plague (e.g., Rogers and Randolph, 2000; Kuhn et al., 2003). A significant relationship between the increase in rainfall and the increased cholera morbidity was recorded in Haiti (Eisenberg et al., 2013). In Bangladesh, the highest incidence of malaria, diarrhoea, and pneumonia was observed during the rainy season (Chowdhury et al., 2018). Research on malaria revealed the impact of interannual climate variability associated with El Niño on health, including
epidemics in India and Sri Lanka (Bouma et al., 1996) and some countries in South America (Poveda et al., 2011).

Although the relationship between climate components, such as temperature, precipitation, humidity, and health status is reasonably well documented, it concerns only recent populations, while there is a gap in research concerning the populations



from the past, although this is crucial in order to analyse long-term developments often associated with climate change. Studies have pinpointed the interrelationship between high temperatures in the summer and excess infant mortality, particularly in
the areas where mothers did not breastfeed their infants (e.g., Vögele, 2010). When it comes to acute infectious diseases, until the end of the 19th century and for the first decades of the 20th century, studies on morbidity and mortality from infectious diseases were limited to elucidating risk/causative factors. The work on the cholera epidemic in the 1850s in the context of ecological conditions in urban space, conducted by John Snow, is considered to be a starting point the infectious disease epidemiology systematic research (Fisman, 2007). Previously, miasmatic theories prevailed, ascribing outbreaks of cholera
and typhoid fever to unspecific germs which came to the surface with the rising groundwater (Vögele, 2001). In this context, rainfall was considered to be a major variable. Such a theory provided the scientific framework for the construction of expensive drainage and sewerage systems in the cities. With the rise of bacteriology towards the end of the 19th century, cholera and typhoid became associated with specific waterborne pathogens and concrete aetiology. As a consequence, adequate central water supply systems became crucial for municipalities. We know from literature that deaths due to some causes used
to show annual seasonal patterns which, to some extent, could have been shaped by climate components, such as temperature or precipitation. A historically observed summer increase in the deaths caused by epidemics of infectious diseases, especially cholera, diarrhoea, dysentery, and typhoid fever was emphasized in the literature previously (e.g., Woods et al., 1988, 1989; Landers, 1993; Liczbińska, 2009a, b, 2011, 2015; Budnik and Liczbińska, 2015).

        Regarding historical populations, according to the best of our knowledge, there are no studies highlighting the
relationship between the risk of death from waterborne and airborne diseases and the characteristics of the local weather/climate, described by temperatures and precipitation patterns. So far, the literature has not been focused on the epidemiology of  the 19th-century Polish urban space in climatological context either. The present study examines the relationship between temperature values and precipitation rates as explanatory variables and the probability of death due to waterborne, airborne diseases and other causes in a historical urban space as responses. We focused on two types
of infectious diseases due to their high burden for the 19th-century populations, and the growing evidence from contemporary populations that they are related to climate/weather. We study the relationship using formalized statistical models reflecting the discrete nature of the response data via multinomial logistic regression.

**2 The city of Poznań in the 19th century**

After the Congress of Vienna in February 1815, the city of Poznań became the capital of the Grand Duchy of Poznań, which was in territorial and administrative terms a part of the Prussian state. The Prussian authorities soon decided to turn the city into a fortress. Fortifications to be built around Poznań were to prevent the threat of the Russian army entering Prussia in case of war. Apart from the very city centre, the area surrounded by the fortifications included the villages of Piotrowo and Berdychowo with their local government, while areas such as St. John's settlement, Zawady, Miasteczko (the present district
of Rataje), Wilda, Gaj, part of Pęcław (Kundorf), Wymykowo, Wenetowo, Stosy, Łazarz and the area of the village of Winiary were left outside of the 19th-century city walls. Poznań was then enclosed within the fortified walls till the beginning





of the 20th century. Turning the city of Poznań into a fortress surrounded by walls inhabited the city's spatial development for long years. The law from 1847, prohibiting any settlement on the city outskirts, acted as an additional inhibition. The area within the fortified walls was mere 50 hectares of land, a considerable part of which was taken over by the military

(Trzeciakowska and Trzeciakowski, 1987; Wędzki, 1994). With time, the city started to suffer from the lack of free space and at the end of the 19th century it was virtually suffocating within the surrounding walls. On top of that, infrastructure was deficient. Confining the city within a small, enclosed area inhibited its demographic growth, and migration was the main factor in deciding the city's demographic dynamics. After 1867, the major barriers inhibiting the influx of population from rural areas disappeared as all previously applicable charges for permanent stay registration were abolished. This triggered a mass

population movement into Poznań. After 1870, the demand for labour force increased even further. Along with industrial development, the city's economic situation began to improve slowly. Poznań became an attractive destination mainly for the population of Great Poland's rural areas looking for jobs in the city. Due to the lack of space in the overcrowded city centre, the incoming migrants settled mainly on the outskirts (Kędelski, 1994). In 1900, a decision was taken to demolish and remove the west belt of fortifications and to incorporate the villages of Jeżyce, Łazarz, Górczyn, and Wilda into the city (Wędzki,

110 1994).

The ecological conditions in Poznań deteriorated increasingly and in parallel with the population increase. Until the mid-1860s, Poznań had no modern central water supply system, and the municipal sewage system was built no sooner than at the end of the century (Trzeciakowska and Trzeciakowski, 1987). Water was drawn from shallow wells or directly from the Warta River (Kaniecki, 2004). High population density and lack of urban infrastructure made the "urban ecology" within the

walls of the fortress really bad. This translated into the health status of Poznań inhabitants, who often suffered from outbreaks of epidemics, i.e., cholera (Piankowski, 1988; Liczbińska, 2021). The mortality toll due to epidemics of contagious diseases, such as cholera, scarlet, typhoid fever, and measles was very high, especially among infants and young children. High temperatures in the summer were conducive to the seasonal increase in the number of deaths due to infectious diseases (Liczbińska, 2015). Infant mortality in Poznań was estimated as >250 deaths per 1,000 live births, and during the intervals

of the epidemics, infant mortality was >300 per 1,000 live births. The exceedingly high infant mortality rates were associated with the low values of life expectancy at birth, i.e., $e_0 = \sim20$ years (Liczbińska, 2009a, b, 2011, 2015).

## 3 Meteorological observations in the 19th-century Poznań

The first systematic air temperature observations in Greater Poland took place after the creation of a measurement network

by the Prussian Meteorological Institute established in Berlin in 1847. The first temperature measurements began in Poznań on January 1, 1848, in the meteorological station at 1 Pocztowa Street (Smosarski, 1925; Kolendowicz et al., 2019). This station operated until March 1862, but in April that year, it was moved to 1 Grobla Street and operated there until September 1867. At the station on Grobla Str., the thermometer was set at 2.5 m above ground level (AGL). In October 1867, the observations were moved to 1 Zielona Str. and continued being conducted there until July 1885. In August 1885, the location

of the station was changed from 1 Zielona Str. to 2 Zielona Str. In that station, temperatures were measured with a thermometer





set at 6.2 m AGL. The station at 2 Zielona Str. operated until September 1892, when it was moved to 3 Długa Street, where the thermometers were placed at 8.6 m AGL. The station on Długa Street operated until December 1911 (Smosarski, 1925). Until 1892, temperature measurements were taken with a thermometer without a cover and placed outside the window. Since September 1892, zinc sheet sheaths were used on thermometers to protect the measurements against atmospheric factors such as wind, sun, and humidity, which distorted the results (Smosarski, 1925). Until 1884, measurements were taken three times a day: at 6 am, 2 pm, and 10 pm. Since December 1884, the hours of observation were changed for 7 am, 2 pm, and 9 pm. This continued until May 1919. Meteorological observations in Poznań were carried out by professors at the Municipal Real School: first by Spiller, and since April 1862 – by Magener. In 1889, meteorological measurements were taken by the Physical Institute of the Royal Academy (Smosarski, 1925). Initially, a Mahlmann rain gauge was used to measure precipitation, which until September 1867 was placed in the yard at 1 Grobla Str., at 2.6 meters AGL. From October 1867 to 1873, the rain gauge was in the garden at 1 Zielona Str., and then it was moved to Pocztowa Str. After 1880, Osnaghi's and then Hellman's rain gauges were introduced into precipitation observations. Those rain gauges were placed on poles at 1 m AGL. The measurements were taken once a day: at 2 pm, and after 1885 – at 7 am. In the 1890s, a meteorological station operated also in Jeżyce, which at that time was a village near Poznań. The thermometers were set at 9.5 m AGL, and the rain gauge was placed in the garden, at 1 m AGL. A pharmacist named Wildt was the manager of the station. Observations were taken three times a day: at 7 am, 2 pm, and 9 pm (Smosarski, 1925).

## 4  Dataset

Data on mortality come from parish death registers for the years from 1850 to 1900, deposited in the State Archives in Poznań. Each individual entry contained the date of death (day, month, year), place of residence, and the cause of death. Records for the following parishes were used: St. Margaret, St. Mary Magdalene, Holy Cross, and St. Martin. For the analysis, 35,005 individual entries containing the cause of death recorded were used. The specific causes of death were categorized as follows: 1) Waterborne diseases (WBD): cholera, typhoid fever, dysentery, diarrhoea, vomiting; 2) Airborne diseases (ABD): measles, smallpox, whooping cough, pulmonary tuberculosis, diphtheria, croup, influenza, measles, scarlet fever, scarlet fever, rash; 3) Others. Therefore, we can consider the three-cause exhaustive death classification as an instance of multinomial distribution. For each deceased person, the following meteorological data were assigned: the average monthly temperatures in the month of death and in the month preceding the month of death (LAG1temperature), and the average rainfall in the month of death and the month preceding the month of death (LAG1 rainfall). The data on temperatures and precipitation for Poznań for the years 1850–1900 were taken from the monograph by Władysław Smosarski: *"Temperature and precipitation in Greater Poland"*, published in 1925 by the Ministry of Religious Affairs and Public Education, in Poznań. As mentioned earlier, the city of Poznań (Posen) was considered by the Prussian authorities an important strategic point in the East, and therefore in 1829, the construction of a ring of fortifications began, which encircled the city on the left and the right bank. Within the fortification ring, there were distinguished five quarters (1–5) (Figure 1). Quarters 1–4 included the area on the left bank of the Warta River, while quarter 5 encompassed the areas located on the right bank of the Warta River (*Polizeipräsidium*





*Posen...*, 1866–1868; Piankowski, 1987). Quarters 1–5 differed in terms of ecological conditions. To each individual deceased person, a quarter identification (quarter number) was assigned, based on his/her address.

## 5 Methods

All analyses are based on multinomial logistic regression (Hosmer and Lemeshow, 2000; Harrell, 2015). We modelled
the probabilities of death due to waterborne diseases (WBD), airborne diseases (ABD), and others as a function of several covariates simultaneously. Modelling in the multinomial logistic regression was conducted with a flexible GAM (Generalized Additive Model; Hastie and Tibshirani, 1990; Wood, 2017) approach, which made it possible to capture both parametric (e.g., linear) and general smooth effects (e.g., penalized-spline-implemented curves). We explored various models and came up with a model structure where the probability of death depended on the year in which it occurred (allowing for
flexible/smooth/nonparametric annual trends – as we do not have any a priori information about the shape of the trends as mathematical functions), one-month-lagged average temperature values and one-month-lagged precipitation rates. It is interesting to note that the one-month lagged models for both temperature and precipitation patterns were always better (in terms of AIC, Akaike Information Criterion;  Burnham and Anderson 2002) than non-lagged models, so we used the lagged version. The covariate effects can be different for different death causes (model allows for cause-specific actions of a concrete
covariate). Due to the well-known identifiability reasons (Hosmer and Lemeshow, 2000), the multinomial model of the three possible response categories (waterborne, airborne, other) has in fact two dimensions and not three, as this third category constitutes a baseline. We chose the "other" category as the baseline. We fitted two modifications of the model – one non-spatial or purely temporal (without distinguishing different city quarters) and the other spatial-temporal (allowing for varying incidence among quarters). As a sensitivity check, the quarter model was then implemented in two ways – as a GMRF
(Gaussian Markov Random Field; Rue and Held, 2005), and in the fixed quarter effects version. Since the results of the two are very similar, we prefer the simpler fixed-effects quarter version and present its results only. The models were fitted in R (R-core Team, 2023), using the mgcv package (Wood, 2017). They were implemented with complexity-penalized splines for smooth components (Harrell, 2009; Wood, 2017), estimated via penalized likelihood with penalization constants being held at their REML (restricted maximum likelihood; Harville, 1977; Wood et al., 2016) estimates.


## 6 Results and discussion

### 6.1 Precipitation and temperature effects on the probability of deaths from WBD and ABD

The detailed parametric and non-parametric effects for monthly precipitation rates and temperature values on the probability of deaths due to waterborne diseases and airborne diseases are summarized in Table 1. All effects in the model are significant,
except for the smooth lagged average monthly temperature effect for airborne diseases. The growing lagged precipitation rates increased the probability of death both from waterborne and airborne deceases. The lagged monthly average precipitation effect was much bigger for deaths from waterborne diseases than for deaths from airborne ones. Moreover, smooth effect of lagged monthly average temperature value was bigger for deaths due to waterborne diseases than deaths caused by airborne



diseases. Similarly, the annual trends were more significant for waterborne diseases than for airborne diseases. The results
from the purely temporal model are presented in Figures 2–3. Note that the lagged average temperature effect is larger for
waterborne diseases and is of different shape than for airborne diseases (Figures 4–5). For waterborne diseases the overall
effect was increasing, although much faster in higher temperatures (approximately above 15°C) than in lower (colder) ones
(Figure 4).

Many epidemiological studies of contemporary populations from different regions of the world have shown that
precipitation and temperature are major environmental factors linked to the prevalence of waterborne diseases (e.g., Singh
et al., 2001; Hashizume et al., 2008; Drayna et al., 2010; Carlton, et al., 2014; Eisenberg et al., 2013; Chowdhury et al., 2018;
Levy et al., 2018; Deshpande et al., 2020). High temperatures and humidity favour multiplication and survival of waterborne
pathogens and their vectors (Singh et al., 2001; Semenza and Menne, 2009; Drayna et al., 2010; Carlton et al., 2014, 2016).
Periods of heavy rainfall  are also related to the prevalence of waterborne diseases by increasing the likelihood of water
contamination from sewages (Moors et al., 2013).They flush faecal material into waterways and into surface drinking water
and enhance the multiplication of intestinal pathogens. Heavy rains increase hydrologic transport-driven iron availability
in environmental waters, leading to increased microbial growth (Faruque et al., 2005), they can flush accumulated pathogens
into surface water directly, through runoff, and indirectly, through the mobilization of bacteria in soil (Carlton et al., 2014).

In the case of airborne diseases, our research showed the temperature effects to be non-monotonic: both increasing
and decreasing (Figure 5). This is probably due to wide temperature spectrum necessary for survival of airborne diseases'
agents. Tang (2009) states that bacteria within the same structural classification (e.g., Gram-negative) may vary in how they
respond to temperature and humidity. Research of Harper (1961) showed that low temperatures, 7–8°C, were optimal for
airborne influenza survival. It was confirmed by Lowen and colleagues (2007), who proved that influenza was transmitted
through the air most readily in cold and dry conditions. Studies have shown that temperatures above about 24°C appear
to reduce the survival of airborne bacteria (Tang, 2009).This has been confirmed by such agents as Gram-negative, Gram-
positive, and intracellular bacteria (Tang, 2009): *Pseudomonas* (Handley and Webster, 1995), *Salmonella* (Dinter and Muller,
1988), *Serratia* (Ehrlich et al. 1970), *Escherichia* (Wathes et al.,1986), *Bacillus* (Ehrlich et al., 1970), *Bordetella* (Stehmann
et al., 1992), *Chlamydia* (Theunissen et al., 1993). As regards fungi, which are also airborne disease agents, there are  studies
confirming their relationship with high temperatures and humidity and also with low winter-temperatures and dry air (Oliviera
et al., 2005).

The lags after rainfall and temperature measurements for waterborne diseases, obtained for the 19th-century Poznań,
have been also observed in contemporary populations, for example, in Hong Kong (Wang et al., 2018), the United States
(Drayna et al., 2010), and New Zealand (Lai et al., 2020). This phenomenon could have been related to the incubation periods
of waterborne pathogens, which vary in length in various types of pathogens. The incubation period of *Vibrio cholerae*
is consistent with the duration of 1–5 days (Azman et al., 2013; Eisenberg et al., 2013). For diarrhoea-causing pathogens,
mainly rotavirus and pathogenic *Escherichia coli*, the incubation period is usually around 1week (Eisenberg et al., 2003), while
lags reported in cholera endemic settings range from several weeks to 2 months (Hashizume et al., 2008, 2010). Bacillary



dysentery symptoms begin within 2 to 10 days of infection while amoebiasis starts more gradually and usually lasts about 2 weeks (*Infectious Intestinal Disease…*, 2012). Meanwhile, *Salmonella typhi* and *Salmonella paratyphi* incubate between

1 to 3 weeks (*Infectious Intestinal Disease…*, 2012). A similar maximum three-week incubation period has *Listeria monocytogenes* (*Infectious Intestinal Disease…*, 2012). In the 19th-century Poznań longer lags in the effects of temperature and precipitation for waterborne diseases could have been caused through the accumulation of bio-organic materials at the bottoms of the water reservoirs, transport-driven with rainfalls from discharged sewage, and then their decay under the high ambient and water temperature. Delayed effects of measured temperatures and precipitation for waterborne diseases could

have also been caused through plankton blooms trigged by nutrients driven into the waters with heavy rains. The city was in a network of three rivers with plankton blooms from the spring through summer months until autumn. Biological material stayed at the bottoms of the rivers until fall, when the temperature was still high (Kaniecki, 2004). The annual sanitary reports on water cleanliness in the Warta River highlighted the contamination with such substances as nitric acid, chlorine, ammonia, and hydrogen sulphide. This did not prevent people from using water from the river for consumption (Kaniecki, 2004). Poznań

at that time suffered from poor sanitation conditions and lack of access to clean water. Heavy rains contaminated surface and groundwater by sewage, which was then used by Poznań inhabitants for drinking, cooking, and washing clothes. The sanitary infrastructure was non-existent practically until the end of the 19th century, favouring the spread of infections with urban sewage. Water was drawn from shallow wells which were often polluted with harmful sewage discharged directly to the moats. There were also street gutters with ineffective drainage full of still, contaminated water during the summer heat periods

(Kaniecki, 2004; Liczbińska, 2011). The sewage system was opened in 1896 (Trzeciakowska and Trzeciakowski, 1987). It was very late comparing to other European cities, such as London (1850s), Paris (1850s), Hamburg (1842), Leipzig (1860), Frankfurt (1867), Zurich (1867) and Prague (1868) (Kappner, 2019). A modern water supply system was completed only in 1866, but it was systematically developed with the increase in the number of inhabitants and dwellings. Although, between 1889 and 1913, it extended from 29 to 176 km (Sobczak, 1979), it was of little significance when compared with Berlin, where

the water supply network was at that time of 661,246 metres (Vögele, 1998). Actually, death rates from waterborne diseases remained high until the end of the 19th century (Liczbińska, 2015).

Airborne agents are transmitted through small particles suspended in the air. Viruses, bacteria, or fungi are the agents spread through breathing, talking, coughing, sneezing, or raising dust (Božič and Kandič, 2021). The dynamics of their survival and spread is influenced by ambient temperature and humidity and favoured more by population density than sanitary and

epidemiological level. Hence, lagged effects of average temperatures for airborne diseases were smaller than for waterborne diseases. Moreover, many airborne agents, such as viruses causing influenza, have a relatively short timescale which resulted in a very short lag. For example, the incubation period for influenza is typically 1–4 days (Belshan at al., 2014).

## 6.2 Spatial (quarter) differences in the probability of deaths from WBD and ABD

In the 19th-century Poznań the death toll wreaked by waterborne and airborne diseases depended on the quarter of residence, which was related to the population density, access to clean water intakes, the level of hygiene, the standard of living





conditions, etc. In our research, when the city quarters were introduced to the model, the effects of average precipitation rates and temperature values on the probability of death due to waterborne and airborne diseases were significant, except for lagged monthly average temperature effect upon airborne diseases (Table 2). This model suggests that there are substantial differences

in the incidence of deaths due to waterborne and airborne diseases in different quarters. The results from the spatial-temporal models are presented in Figures 6–11, while parametric coefficients for monthly rainfall rates and direct effects of city quarters on the probability of deaths due to waterborne diseases (WBD) and airborne diseases (ABD) in Table 3. All effects are statistically significant. The smooth (Figures 6–7) and lagged (Figures 8–9) effects are very similar to those from the previous models.

275         Figure 10 demonstrates spatially varying probability of death due to waterborne diseases in 5 city quarters. In quarter 1 probability of deaths caused by waterborne diseases was higher than in quarter 2, while in quarter 3 it was higher than in quarter 4, where in turn it was lower than in quarter 5. Quarters 1 to 4 were enclosed within the fortress walls, while quarter 5 was located on the right bank of Warta River, i.e., outside it (see Figure 1). In the light of police reports (*Polizeipräsidium Posen…*,1866–1868) in 1866 almost 83% of the city inhabitants lived within the fortress (quarters 1–4), while the rest in

quarter 5. At the end of the 19th century there were almost 8,000 inhabitants per 1 $km^2$ enclosed within the walls (Kruszka, 2004). In such an overcrowded space there was a great epidemiological problem related to the lack of sanitary infrastructure and clean water intakes. A street was at the same time a gutter, and every water reservoir became an intake for drinking water. Since underground waters were shallow, they were heavily contaminated by leaks from cesspits and sewers. Waste was poured onto the streets or courtyards, from where it flowed down the sewers to the wells. Until the mid-19th century, water was drawn

mainly from the rivers, shallow wells, intakes located near the cemeteries, and from shallow ditches supplied with water from ponds. The first wooden waterworks were built in 1847. They drew water from the spring in Winiary and distributed it to the wells at Świętokrzyski Square, Bernardyński Square and to the suburb of St. Adalbert. The total length of that aqueduct was 2.5 km and it was insufficient to supply all inhabitants with clean water (Kaniecki, 2004).

        Inhabitants of quarters: 1, 2 and 5 had access to the Warta River, being a potential source of drinking water for many

of them. In those sectors the highest mortality due to waterborne diseases was observed. However, the Market Square area and neighbouring streets (quarter 1) were the only places in the city with the access to potable water. It was related to the fact that in the 16th century, the fountains with fresh spring water supplied by pipelines from outside the city were built there (Kaniecki, 2004). The surplus of deaths from waterborne diseases in quarter 1 (privileged in terms of the access to clean water) and quarter 3 (with no direct access to the rivers) was related to the presence of two hospitals in those quarters. The first was established

in the 1820s in the former Bernardine Convent (in quarter 1), and the second was the city hospital on Szkolna Street, opened in 1854 (in quarter 3) (Kaniecki, 2004). In the poor working-class district, located on the right bank of the Warta River, encompassing such districts as  Chwaliszewo, Śródka, Zawady and Ostrówek (quarter 5) drinking water was taken straight from the rivers, moats, and ponds (Kaniecki, 2004).  Figure 11 demonstrates spatially varying probability of death due to airborne diseases in Poznań: the highest in quarter 5, and the lowest in quarter 4, while it was quite similar in quarters 1–3.

Just as mortality from waterborne diseases depends mainly on the level of hygiene, the mortality from airborne diseases was





influenced by many other factors: environmental conditions, such as population density, standard of living, nutrition, but also individual characteristics of the immune system (Liczbińska, 2009a).The biological standard of living could have been the worst in quarter 5. This district was inhabited by the poorest social strata: small craftsmen and workers, whose wages did not ensure the subsistence level, workers, labourers, and daily-labourers (Trzeciakowska and Trzeciakowski, 1987). While

the wages of skilled workers were sufficient to maintain a very basic standard of living, the wages of unskilled workers did not ensure their minimum subsistence (Liczbińska, 2015). Workday in overcrowded industrial plants became increasingly longer and working conditions deteriorated. In many craft workshops, the twelve-hour workday was often extended to 16 hours (Łuczak, 1965). Still in 1900, nearly 45% of the Poznań population lived in one-room dwellings, often cramped, damp and unheated, one room being sometimes shared by 5 to 12 people (Łuczak, 1965), which facilitated the spread of airborne agents

in the overcrowded space. Poor health condition of workers deteriorated due to malnutrition affecting their immune systems (Munteanu and Schwartz, 2022; Wu et al., 2019). The city within the fortress was inhabited by the wealthier part of Poznań society: artisans of various trades, including shoemakers, tailors, bricklayers, and carpenters. A certain professional group was represented by the owners of taverns, restaurants, wine bars and beer halls located in the Market Square. The intelligentsia (doctors, teachers, officials, etc.) also lived in the city centre (quarters 1 to 4) (Trzeciakowska and Trzeciakowski, 1987).


**7 Conclusion**

Individual demographic data on the causes of death provide a deep insight into the relationship between mortality and its various ecological and social determinants. The extensive body of literature has highlighted that poor urban ecology in the 19th-century populations was related to the lack of sewage system and clean water intakes as the main factors responsible for

death surpluses. Meanwhile, understanding the impact of meteorological phenomena on infectious diseases in past populations is of key importance in studying their health status and adaptation to environmental conditions. In our study, using monthly average temperature values and precipitation rates, we demonstrated the influence of climate on the probability of death from infectious diseases. We pointed out the complexity of the relationship between precipitation rates and air temperature values and waterborne and airborne diseases in city districts that differed in terms of hygiene and living standards. Our findings

underscore the need to account for climatic factors when evaluating the probability of death due to infectious diseases in urban spaces. Moreover, further research is needed to elucidate to what extent socioeconomic and biophysical factors can modify climate-disease relationships. While the burden of infectious diseases has been declining since the second half of the 19th century as a result of improved hygiene, medical progress, and improved quality of life, climate change may have slowed the progress in reducing the burden of these diseases.


**Data availability.** The data on mortality were collected within the project supported by the National Science Centre, Poland, under Grant *Cholera epidemic in 1866 as a turning point in the history of Poznań,* No. UMO-2021/41/B/HS3/00594. Data on temperatures were drawn from already published monograph of Władysław Smosarski, *Temperature and precipitation in Greater Poland*, published in 1925 by the Ministry of Religious Affairs and Public Education, in Poznań.





Currently all data are stored by Grażyna Liczbińska.

**Author contributions.** GL, MB, JV designed the study. MB provided the statistical analysis and prepared figures 2–11. GL
provided cartographical figure 1. GL and JV provided the historical analysis. All authors discussed the methods and results
and commented on the paper.


**Competing interests.** The contact author has declared that neither they nor their co-authors have any competing interests.

**Financial support**

This work was supported by the National Science Centre, Poland, under Grant *Cholera epidemic in 1866 as a turning point in*
*the history of Poznań,* No. UMO-2021/41/B/HS3/00594.

MB's work has been partially supported by the long-term strategic development financing of the Institute of Computer Science
(Czech Republic RVO 67985807).

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





Table 1. Parametric and nonparametric effects for monthly precipitation rates and temperature values on probability of deaths due to waterborne diseases (WBD) and airborne diseases (ABD)

| Summary parametric effects | | | |
|---|---|---|---|
| Effect name | Estimate | Standard error | z-value |
| Intercept (WBD) | -2.517 | 0.039 | -63.768 |
| Lag 1 monthly precipitation (WBD) | 0.005 | 0.0007 | 6.866 |
| Intercept (ABD) | -1.124 | 0.026 | -43.389 |
| Lag 1 monthly precipitation (ABD) | 0.001 | 0.0005 | 2.325 |
| Summary of nonparametric effects | | | |
| Effect name | Equivalent degrees of freedom (effect complexity) | Ref.df | Chi.sq |
| Annual trend (WBD) | 8.857 | 8.999 | 1492.51 |
| Lag 1 monthly temperature (WBD) | 8.045 | 8.754 | 1408.99 |
| Annual trend (ABD) | 6.431 | 7.577 | 199.53 |
| Lag 1 monthly temperature (ABD) | 3.361 | 4.206 | 7.944 |






Table 2. Parametric and nonparametric effects for monthly precipitation rates, temperature values and city quarters on probability of deaths due to waterborne diseases (WBD) and airborne diseases (ABD)

| Summary parametric effects | | | |
|---|---|---|---|
| Effect name | Estimate | Standard error | z-value |
| Intercept (WBD) | -2.523 | 0.039 | -63.772 |
| Lag 1 monthly precipitation (WBD) | 0.005 | 0.0007 | 6.484 |
| Intercept (ABD) | -1.129 | 0.026 | -43.580 |
| Lag 1 monthly precipitation (ABD) | 0.001 | 0.0005 | 2.385 |
| Summary of nonparametric effects | | | |
| Effect name | Equivalent degrees of freedom (effect complexity) | Ref.df | Chi.sq |
| Annual trend (WBD) | 8.957 | 8.999 | 1473.049 |
| City quarter (WBD) | 3.647 | 4.000 | 52.106 |
| Lag 1 monthly temperature (WBD) | 8.054 | 8.758 | 1413.774 |
| Annual trend (ABD) | 5.401 | 6.549 | 135.340 |
| City quarter (WBD) | 3.736 | 4.000 | 100.579 |
| Lag 1 monthly temperature (ABD) | 3.311 | 4.145 | 7.313 |





Table 3. Parametric coefficients for city quarters and monthly precipitation rates on probability of deaths due to waterborne diseases (WBD) and airborne diseases (ABD)

| Summary parametric effects | | | |
|---|---|---|---|
| Effect name | Estimate | Standard error | z-value |
| Quarter 1 | -2.349 | 0.049 | -47.959 |
| Quarter 2 | -2.701 | 0.077 | -35.096 |
| Quarter 3 | -2.413 | 0.065 | -37.322 |
| Quarter 4 | -2.696 | 0.052 | -51.871 |
| Quarter 5 | -2.516 | 0.062 | -40.344 |
| Lag 1 monthly precipitation (WBD) | 0.005 | 0.001 | 6.848 |
| Quarter 1 | -1.103 | 0.033 | -33.357 |
| Quarter 2 | -1.144 | 0.048 | -23.598 |
| Quarter 3 | -1.086 | 0.043 | -25.034 |
| Quarter 4 | -1.303 | 0.034 | -37.927 |
| Quarter 5 | -0.896 | 0.039 | -22.925 |
| Lag 1 monthly precipitation (WBD) | 0.001 | 0.000 | 2.389 |



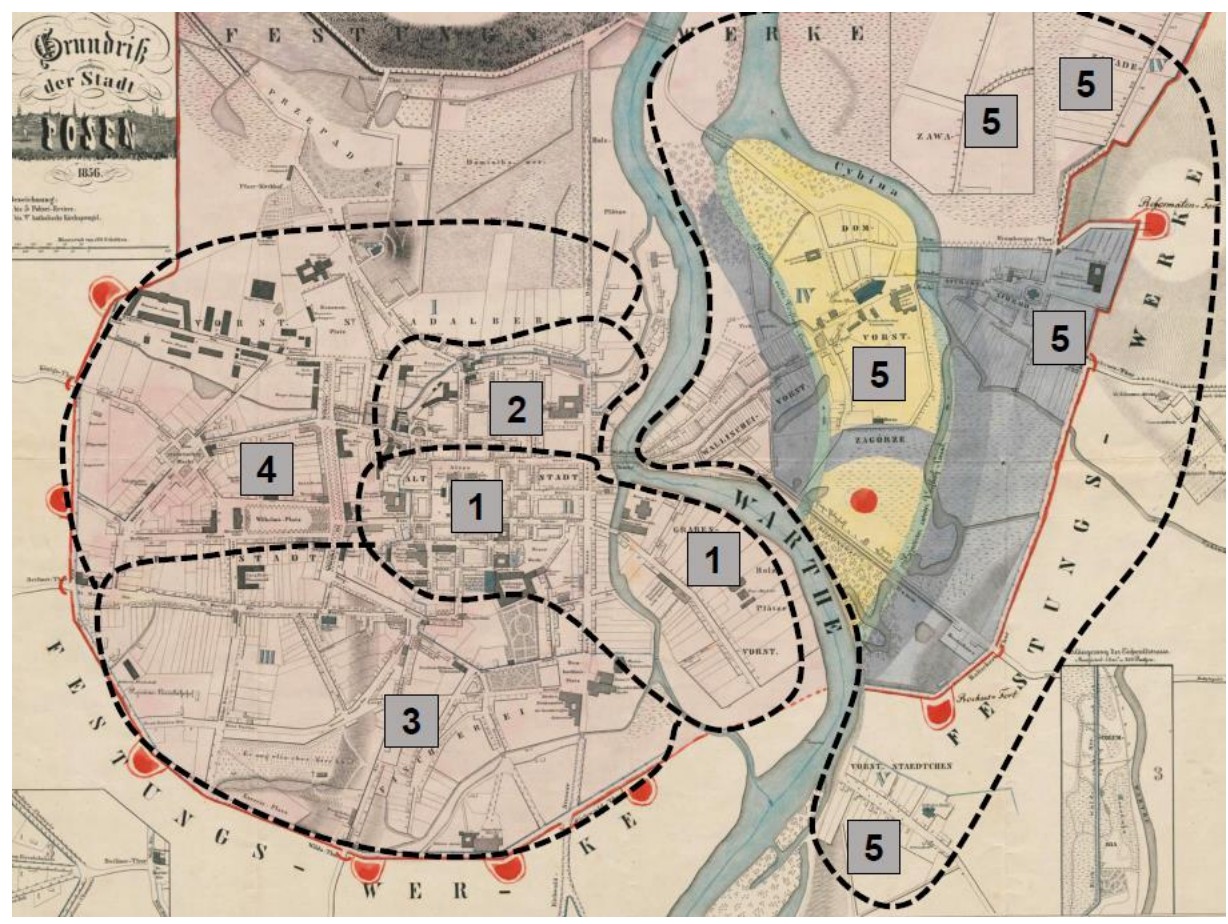

**Figure 1.** City of Poznań divided into five quarters. Map made by Grażyna Liczbińska based on data collected while writing the article and file from https://cyryl.poznan.pl/kolekcja/plany-poznania-do-1945-roku-biblioteka-raczynskich/

(Last access: 19 March 2023).





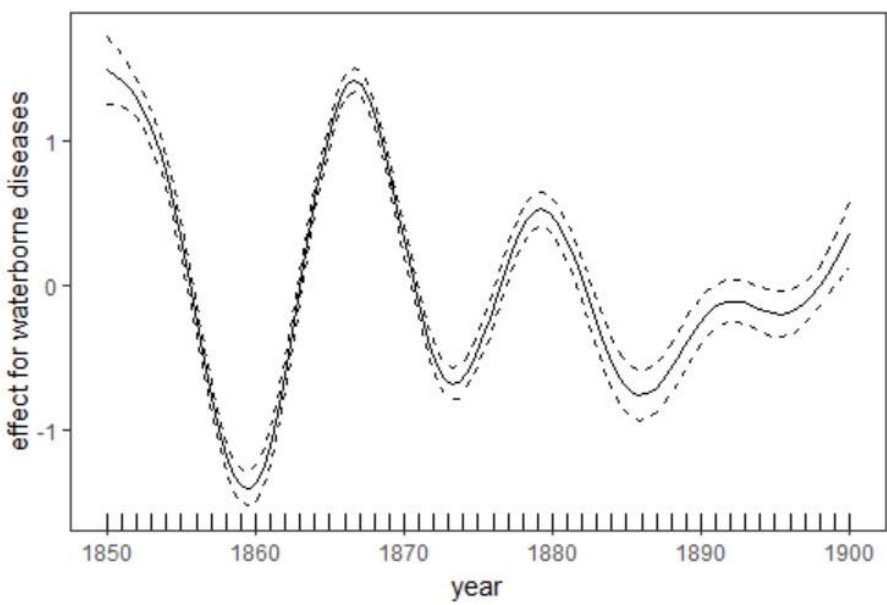

**Figure 2.** Smooth trends (smooth curves and their 95% asymptotic pointwise confidence limits) for waterborne diseases. Figure made by Marek Brabec based on data collected within the project of the National Science Centre, Poland, Grant:

*Cholera epidemic in 1866 as a turning point in the history of Poznań,* No. UMO-2021/41/B/HS3/00594.







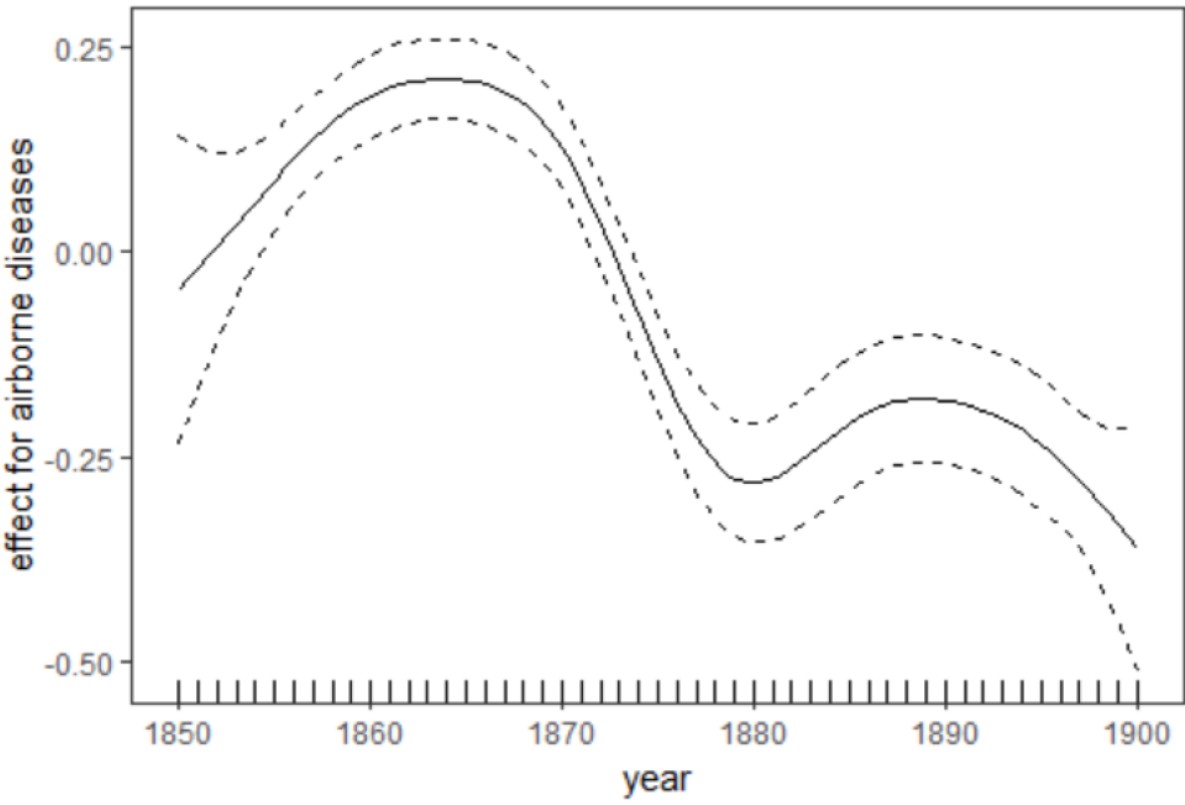

**Figure 3.** Smooth trends (smooth curves and their 95% asymptotic pointwise confidence limits) for airborne diseases. Figure made by Marek Brabec based on data collected within the project of the National Science Centre, Poland, Grant: *Cholera epidemic in 1866 as a turning point in the history of Poznań,* No. UMO-2021/41/B/HS3/00594.





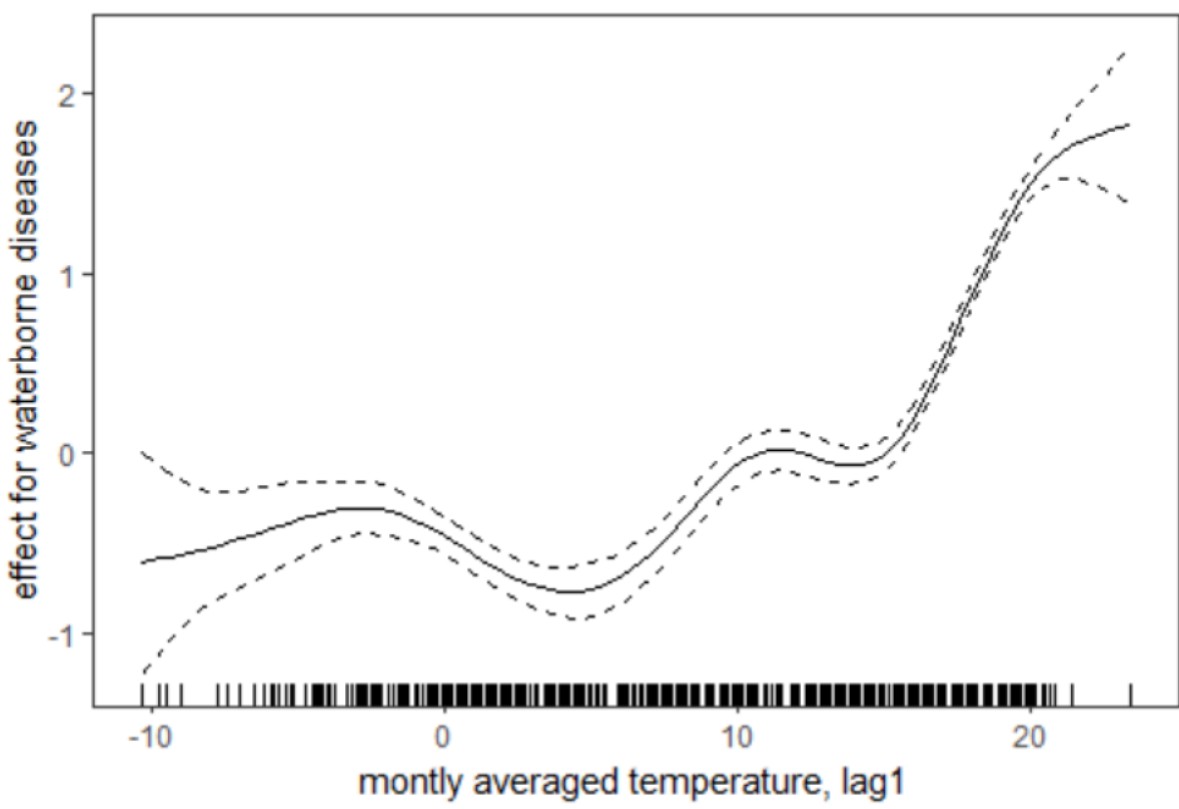

**Figure 4.** Lagged mean temperature effect for waterborne diseases. Figure made by Marek Brabec based on data collected within the project of the National Science Centre, Poland, Grant: *Cholera epidemic in 1866 as a turning point in the history of* 665 *Poznań,* No. UMO-2021/41/B/HS3/00594.







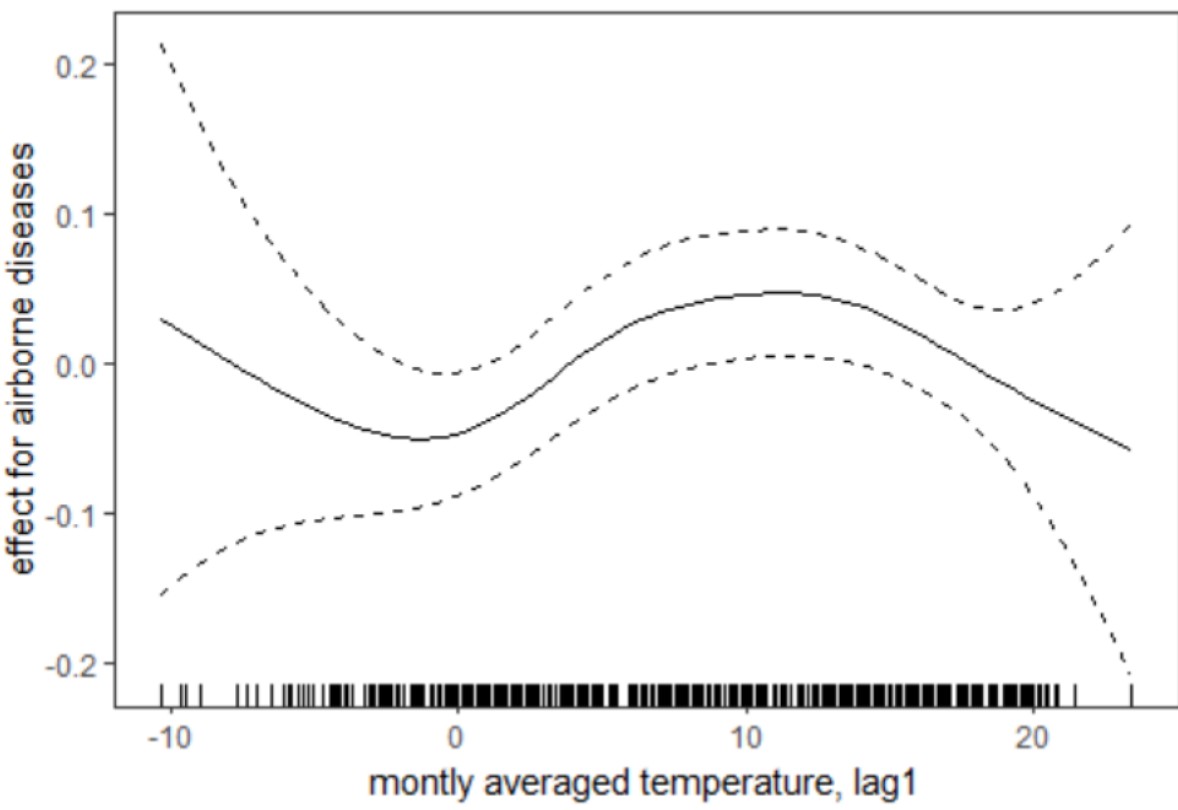

**Figure 5.** Lagged mean temperature effect for airborne diseases. Figure made by Marek Brabec based on data collected within the project of the National Science Centre, Poland, Grant: *Cholera epidemic in 1866 as a turning point in the history of Poznań,* No. UMO-2021/41/B/HS3/00594.








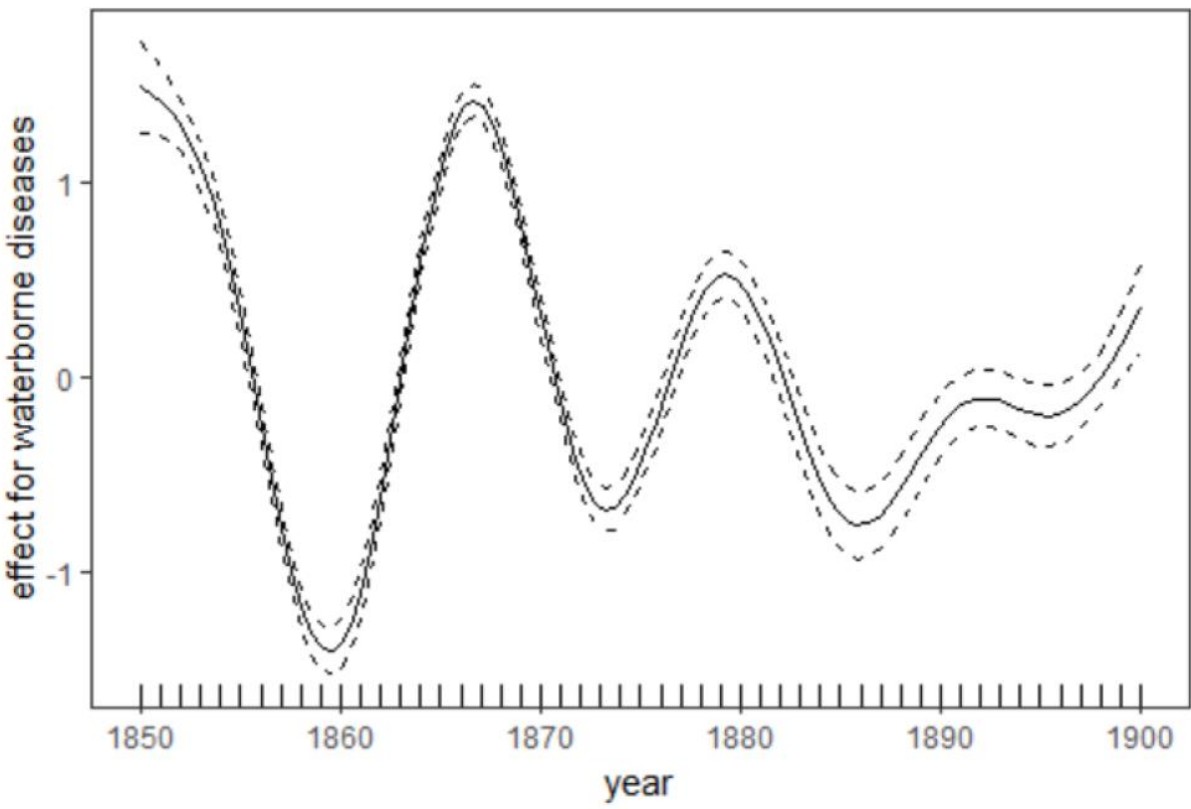

**Figure 6.** Smooth trends of temperature effect for waterborne diseases in spatial-temporal models. Figure made by Marek
Brabec based on data collected within the project of the National Science Centre, Poland, Grant: *Cholera epidemic in 1866 as
a turning point in the history of Poznań,* No. UMO-2021/41/B/HS3/00594.






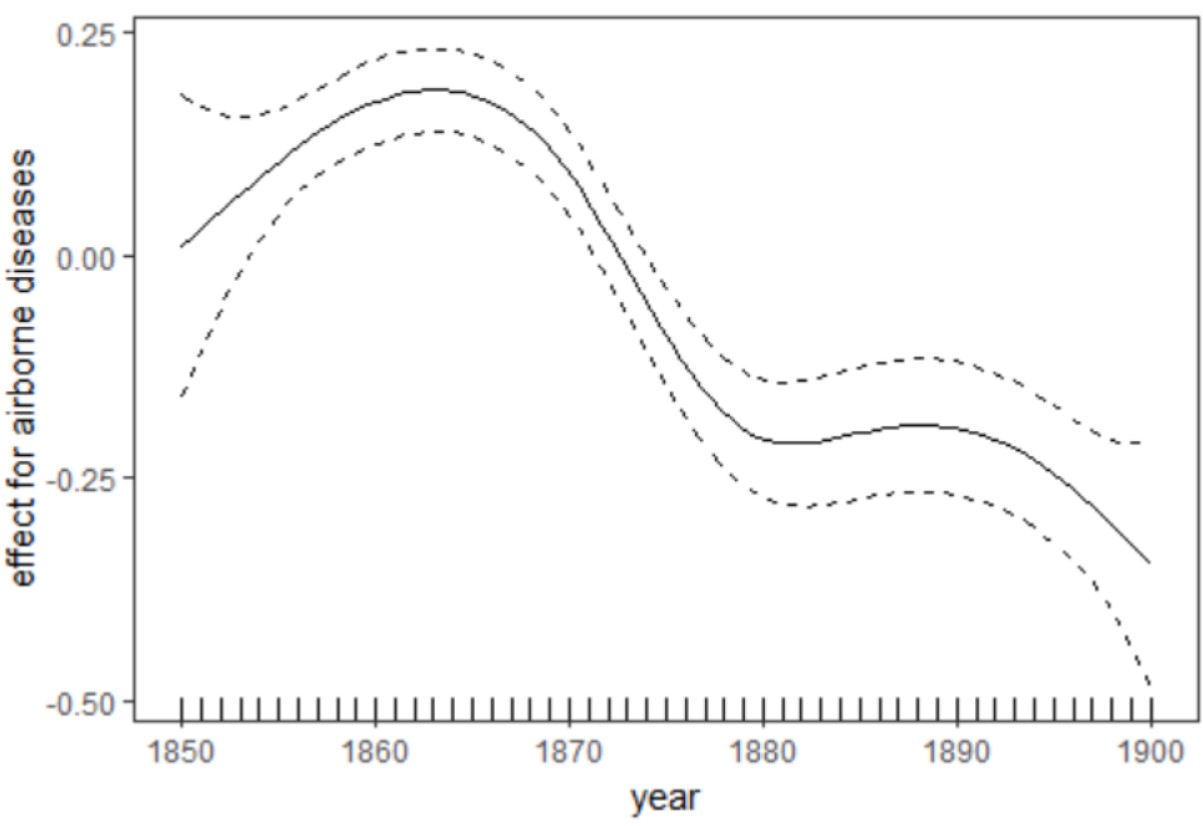

**Figure 7.** Smooth trends of temperature effect for airborne diseases in spatial-temporal models. Figure made by Marek Brabec based on data collected within the project of the National Science Centre, Poland, Grant: *Cholera epidemic in 1866 as a turning point in the history of Poznań,* No. UMO-2021/41/B/HS3/00594.







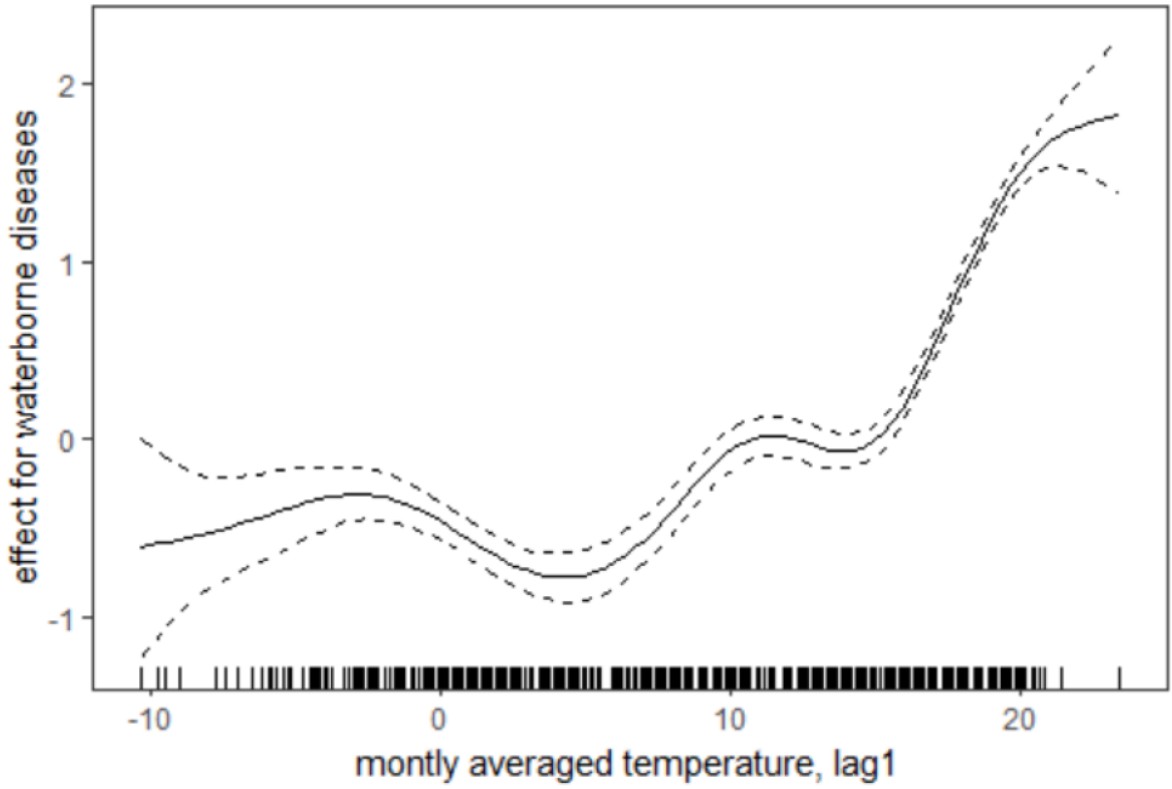


**Figure 8.** Lagged mean temperature effect for waterborne diseases in spatial-temporal models. Figure made by Marek Brabec based on data collected within the project of the National Science Centre, Poland, Grant: *Cholera epidemic in 1866 as a turning point in the history of Poznań,* No. UMO-2021/41/B/HS3/00594.








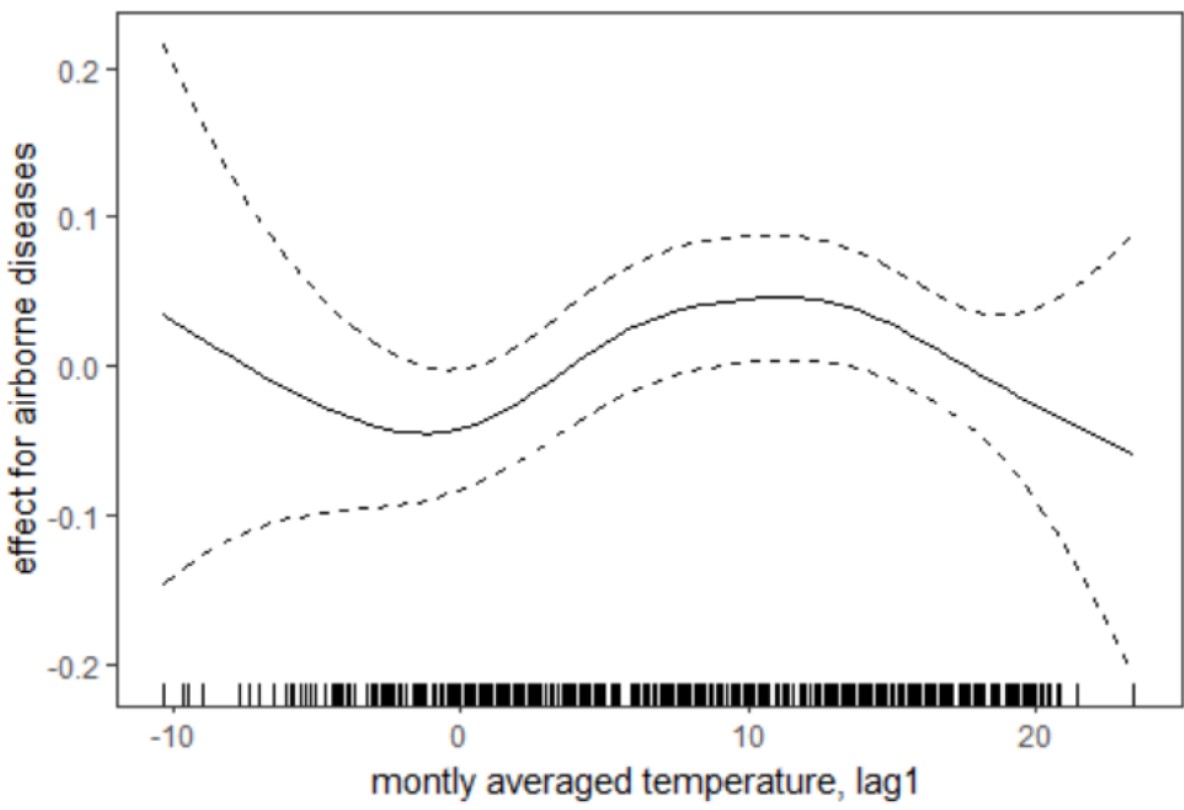

**Figure 9.** Lagged mean temperature effect for airborne diseases in spatial-temporal models. Figure made by Marek Brabec based on data collected within the project of the National Science Centre, Poland, Grant: *Cholera epidemic in 1866 as a turning point in the history of Poznań,* No. UMO-2021/41/B/HS3/00594.








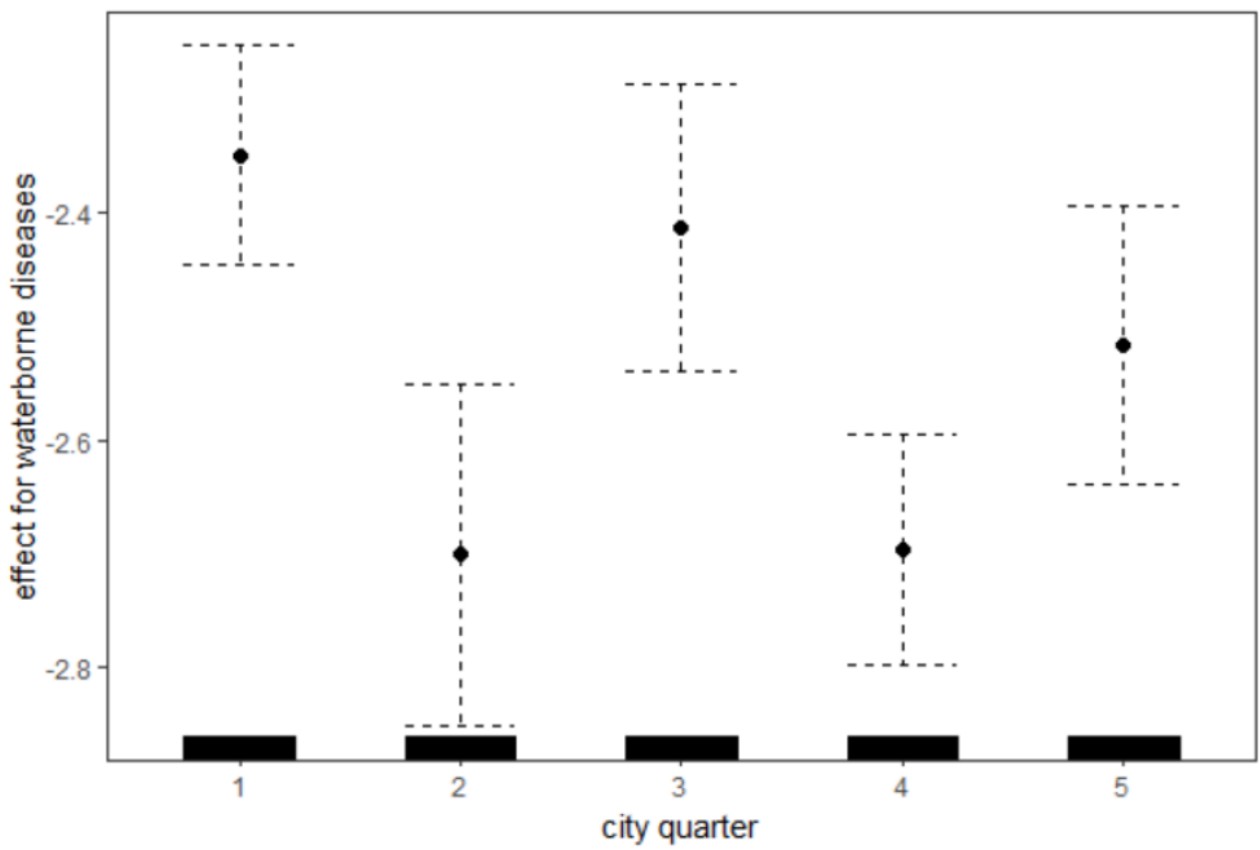

**Figure 10.** Spatially varying probability of death due to waterborne diseases in Poznań quarters. Figure made by Marek Brabec

based on data collected within the project of the National Science Centre, Poland, Grant: *Cholera epidemic in 1866 as a turning*

*point in the history of Poznań,* No. UMO-2021/41/B/HS3/00594.







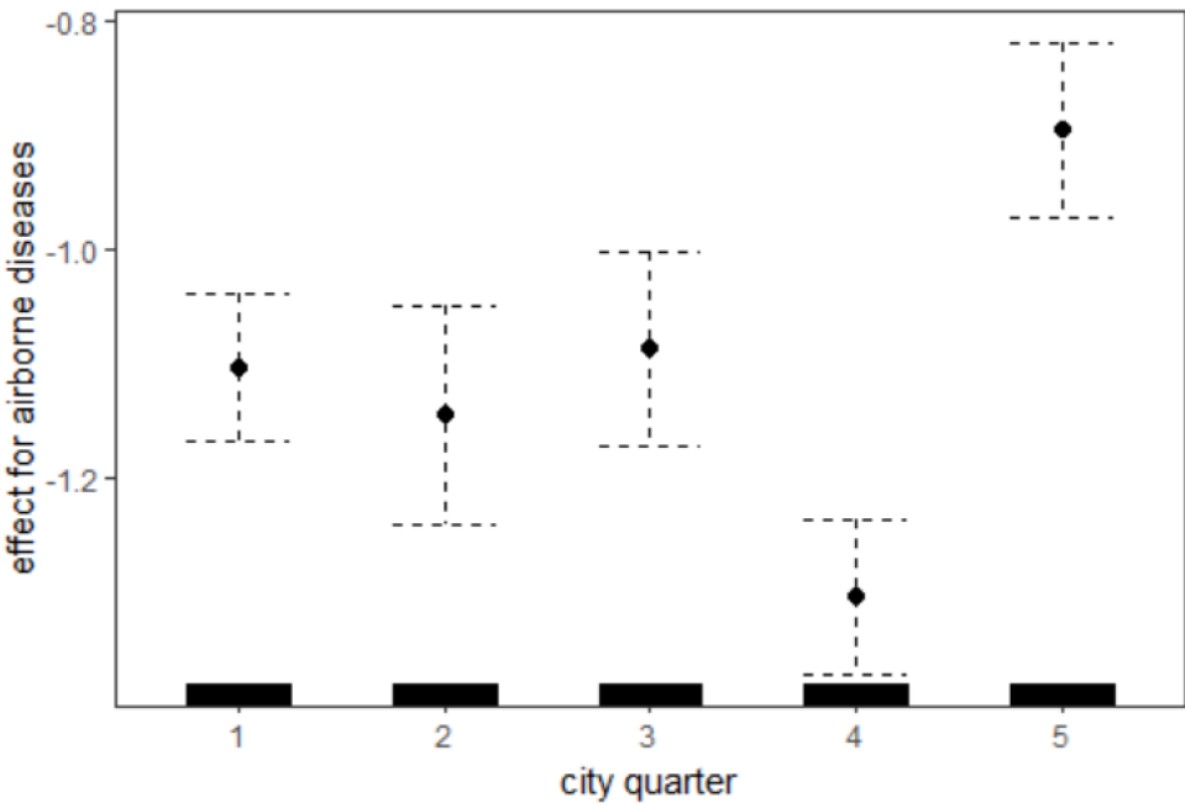

**Figure 11.** Spatially varying probability of death due to airborne diseases in Poznań quarters. Figure made by Marek Brabec

based on data collected within the project of the National Science Centre, Poland, Grant: *Cholera epidemic in 1866 as a turning point in the history of Poznań,* No. UMO-2021/41/B/HS3/00594.