# Peer review of "century Poznań, Poland"

_Climate of the Past, 2023_

## Author Comment (AC3)

**REVIEW 1**

The article "Climate and Disease: the connection between temperature values and precipitation rates and the probability of death due to waterborne and airborne diseases in historical urban space (the evidence from Poznań, Poland)" by Liczbińska et al. is an important contribution to the growing number of European historical climate–health studies. It is especially welcome as most, to date, concerns regions of western Europe rather than central Europe. Indeed, a disproportionate number concerns conditions in the United Kingdom that, due to an early Industrialisation, is far from representative for 19[th] century Europe as a whole.

Thank you.

**Major comments:**

The article focus too much, in its Introduction, on climatic health treats arising from high temperatures. There are numerous health threats, especially in the past, also arising from cold temperatures. The treat of cold temperatures should, in my opinion, be given an equal weight to treat of warm temperature in the background section. Especially in the historical climate–health studies literature, numerous examples of increased mortality following anomalously cold condition can be found that are presumably also highly relevant for Poland. Some suggestion of references to include are found below. I think the article needs to be better placed in a larger research context as many key works appears to be missing from the reference list.

Thank you very much for the review and for this suggestion. In the paper we have focused on high temperatures since they are associated with epidemics of waterborne diseases. Meanwhile in the case of airborne diseases, the relationship with high temperatures is not so obvious. To focus on cold waves/ low temperatures in the Introduction is justified and we will do it in the review. We can also refer to our results, interpreting the influence of lower temperatures on the probability of death due to waterborne and airborne diseases. All literature suggestions provided by the Reviewer will be added in the review.

In addition, I am a bit sceptical that that the living conditions in 19[th] century Poznań really were as poor as suggested by Liczbińska et al. This should be critically discussed as there is a huge disagreement among (economic) historians about the standards of living among the working poor in different parts of pre-industrial and early industrial Europe. Poland, admittedly, had relatively low standards of living compared to northwestern Europe, but I'm still not convinced that they actually were as poor as the article gives the impression of. To cite modern economic historical literature would be helpful to support the statements.

The creation of a fortress surrounded by walls significantly inhibited Poznań's spatial development. The city gradually felt the lack of free space, and by the end of the 19th century it was suffocating in the surrounding walls. At that time, most European cities were experiencing a period of urban development and dynamic modernization. Poznań by contrast suffered from shortages in terms of infrastructure: until the mid-1860s, it did not have a modern water supply system, and only at the end of the century a sewage system was constructed. The city was also often plagued by floods of the Warta River and epidemics. The overcrowded city, with very dense buildings, also posed a risk of fires. In 1838, the authorities issued regulations regarding development, fire safety and health conditions. The technical and sanitary standards of the streets also left much to be desired. Only in the mid-19th century did road investments begin, sidewalks were laid, and gutters were built. The second half of the 19th century was

difficult for the city. Famine, echoes of wars fought in Europe, epidemics, crop failures and elementary disasters led to a decline in the population size. In 1863, there was a significant outflow of Poznań inhabitants to the Kingdom of Poland (Polish lands under the Russian partition) where they participated in the January Uprising (1863). The population growth in 1850 was 0.7 per 1,000 inhabitants, while in 1899, it was 6.1 per 1,000 inhabitants only. In the years of the most severe cholera epidemics, i.e., in 1852, 1855, and 1866 the population growth was at the level of -41.7, -7.83, and -17.9 per 1,000 inhabitants, respectively. The values of population growth above 10 per 1,000 was recorded only after 1900. Migrations were the main factor dynamizing the city's demographic dynamics. After 1867, the barriers preventing the influx of people from the countryside and all fees for permanent registration were abolished and the demand for labour increased. With the development of industry, there was a slow revival of the city's economic activity. Poznań became an attractive destination for people looking for work, mainly from villages of the Greater Poland region. Due to the lack of free space in the already overcrowded City Centre, the migrants settled on the outskirts of the city. The poorest and most dilapidated district of Poznań was the one located on the right bank of the Warta River. The population there lived mostly in overcrowded tenant house outbuildings under undignified conditions. The level of sanitation was very poor in these areas. For example, water was drawn from shallow wells, which were often tainted with harmful sewage from cesspits, gutters, and rubbish sites. During the cholera epidemic of 1866, the Case Fatality Rate in the city was 59%, which means that 59% of people infected with cholera had a fatal outcome, while only 41% recovered.

Craftsmen and white-collar workers predominated in the fortress, while unskilled workers outside it. The latter constituted the poorest part of Poznań society. The salary of unskilled workers ranged 500- 600 marks per year: a man earned 1.6 marks a day, a woman 1.0 marks. These salaries did not provide them with a minimum subsistence. After paying the rent, a worker had less than 1 mk per day to support the whole family. The illiteracy rate was 86%.

**Minor comments:**
Title: The title is far too long. It needs to be shortened, preferably by half.
Our proposal is: **Climate and Disease in urban space (the evidence from 19[th] c. Poznań, Poland)**

Line 13: This part of the sentence does not read well. Furthermore, "other diseases" is too vague.
We agree with the Reviewer's remark. A better solution would be to replace third category "other diseases" with " other causes of death" and provide examples.

Lines 12, 16, 39 and other places: Better to write "temperature levels" and "precipitation amounts".
Thank you for your suggestions. They will be implemented.

Line 16: In English, "block" rather than "quarter" are normally used in this context.
The suggestion will be implemented.

Lines 25–29: I would consider another introduction. The climate adaptation is, besides genetic (which takes a very long time), also social and cultural (and to some extent

even socio-political). If the introduction is not changed this needs to be more clearly stated.

Thank you for this suggestion. We fully agree with this remark. The Introduction section will be re-edited, and changes will be implemented according to the Reviewer's suggestion.

Line 53: Not only warm temperature. Weather conditions are also shaped by cold temperatures.

We agree with the Reviewer's suggestion. This part will be re-edited.

Both high and low temperatures have a significant effect on health. In our work we have received effects of lagged temperature on waterborne and airborne diseases (Figs 8 and 9, respectively). We did not receive low temperature effect on waterborne diseases (fig 8), as was expected (low temperature; left part of Figs 8 and 9).

The probability of deaths due to waterborne disease generally increases with lagged monthly average and at heigh temperatures, above 15°C, it increases even more quickly. there are no surprises at low temperatures (the probability of the waterborne diseases is the lowest in the lowest temperatures, as expected a priori). On the other hand, for airborne diseases (Fig 9), we received the highest probability at, say middle temperatures (local maximum of the effect curve at around 12 centigrade – when the temperatures are not too low and not too high, probably mostly Fall and Spring). Then the probability decreases as the temperature increases. When the temperature decreases from 12°C downwards, the probability of deaths due to airborne disease first decreases (and decreases to local minimum at around the freezing point), with further decrease of temperature, the airborne disease probability tends to increase, but since there were not so many data with very low temperatures (not so many extraordinarily cold years), the super-cold temperature effect is estimated with very high uncertainty (very wide 95% confidence intervals about the estimate) to be safely interpretable.

Lines 56–58: Not really relevant even to cite as the present article concerns Europe.

We agree with this suggestion; it should be removed.

Line 62: Please consider to use another word than "reasonably" here.

We agree with this suggestion; it will be re-edited.

Lines 112–114: I would like to see more exact numbers here.

Poor urban ecology influenced mortality level. In Poznań Infant Mortality Rates were very high to the end of the 19th century. Particularly high IMRs, amounting to more than 300 deaths per 1,000 live births, were observed here in the 1850s-1870s, when Poznań was hit by epidemics of infectious diseases, including cholera in 1852, 1855 and 1866. A gradual improvement in city infrastructure was noted in the 1890s, after the construction of the sewage system, and finally in 1900 when the fortification walls were demolished, which ultimately improved ecological conditions in the city. It is worth mentioning that in the 19th century infectious diseases hit Poznań very often. Particularly dangerous were epidemics of cholera, repeating in 1831, 1837, 1848, 1852, 1855 and 1866. They caused massive and rapid changes in population numbers within a short period of time. There were also outbreaks of measles (in 1857, 1861 and 1869), scarlet fever (1863 and 1874), and smallpox (1871). Life expectancy at birth fluctuated over time and in the years 1875-1884 and 1885-1894 was below 30 years: 29.8 yrs., 28.6 yrs., respectively. A significant improvement was observed in the

decades: 1895-1904 and 1905-1913, 40.03 yrs. and 39.5 yrs., respectively (authors' calculations based on Prussian Statistical Yearbooks).
The city often suffered from Warta floods. In the 19th century the city saw the swollen Warta waters 16 times, 10 events being great floods, of which as many as 7 were classified as of 3rd or 4th category. They rose to even the most elevated sites in Poznań located more than 57 metres above sea level, while the river discharge amounted to 1,000 m3/s or more (e.g., 1,720 m3/s during the flood lasting from 4 March to 30 April 1855 (Kaniecki 2004).

Line 218: Different citation style than elsewhere.
This will be re-edited.

Line 223: I think "fungi" is too vague. There exists many different type of fungi.
Our intention was to list pathogens responsible for infectious diseases, i.e., infectious agents including microorganisms such as bacteria, viruses, fungi, and prions. We did not specify pathogenic fungi in this context.

Line 280: The number of inhabitants per square kilometre is not adding up with other information about the same thing in the article. Please, double-check this information.
We have written that "at the end of the 19th century there were almost 8,000 inhabitants per km2 enclosed within the walls" (line 280). In 1896, the area of the city was 943.4 ha, i.e., almost 9.5 km2. At that time the city had a population of 73,293 (After Galloway, Patrick R., 2007, "Galloway Prussia Database 1861 to 1914"). Our calculations demonstrate that there were over 7,700 people per 1 km2. Our calculations are similar to the numbers provided by Statistics for the City of Poznań (ed. by Kruszka 2008; p. 53), in the light of which there were 7,700-7,800 people per 1 km2 within the fortress. Since the area of the city had not changed until 1900, in the 1860s there were more than 5,600 people per km2, and in the 1870s - more than 6,400 people per km2. Up to 12 people lived in one room.
Historians emphasize that the city was "virtually suffocating within the walls". After the fortress was demolished in 1900, suburban quarters were incorporated into the city. The urban area expanded to 3,300 hectares and the population density decreased to 3,500 per km2.

Line 304 and other places: Please provide a distinction between "workers" and "labourers".
The term worker is a very broad definition comparing with the term labourer. According to Britannica Dictionary, a word "worker" is referred to a person who does a specified type of work or who works in a specified way. A word "labourer" refers to a person who does unskilled physical work: a daily labourer; according to Longman Dictionary, someone whose work needs strength rather than skill.

Lines 308–309: I am highly sceptical too that the conditions were so bad (except in exceptional circumstances).  If this indeed is true – which I do not entirely rule out – it should be supported by more modern (post-Communistic period) scholarship.
Apart from information on hygiene, sanitation, and ecological conditions, provided as a response to earlier comments, we would like to add some information on working conditions in the city. The working day lengthened and conditions in factories deteriorated along withe the city industrialization. In many artisans workshops a twelve-hour working day, often extended to 16 hours, was in force practically until the outbreak

of the First World War. Health and safety in industry and transport left a lot to be desired. Many factories and workshops failed to install protective devices, thus causing an increase in the number of accidents at work. Poor living conditions and malnutrition were additionally to blame for the deteriorating poor health condition of employees. In 1900 nearly 45% of the Poznań population lived in one-room dwellings, often cramped, damp and unheated, one room being sometimes shared by 5 to 12 people. The modern water supply system, opened in 1866, effectively contributed to the improvement of sanitary conditions in the city and was systematically developed with the increase in the number of inhabitants and dwellings. In the years 1889-1913 its length grew from 29 to 176 km. Even so, this was very little compared with Berlin, where the water supply network was nearly six times longer (661,246 metres, Vögele 1999).

All the graphs: They needs to be improved and provided as proper vector graphics. The resolution is very poor. There size and dimensions also need to be streamlined. The font size in the graphs is, in general, too large.
Graphs will be changed in the review.

**Suggested references:**
Thank you. We are going to implement this literature in the review.

Åström, D. O., Edvinsson, S., Hondula, D., Rocklöv, J., and Schumann, B.: On the association between weather variability and total and cause-specific mortality before and during industrialization in Sweden, Demographic Research, 35, 991–1010, 2016.
Bradley, L.: An enquiry into seasonality in baptisms, marriages, and burials, Local Population Studies, 4, 21–40, 1970.
Calleja-Agius, J., England, K., and Calleja, N.: The effect of global warming on mortality, Early Human Development, 155, 105 222, 2021.
Castenbrandt, H.: A forgotten plague: dysentery in Sweden, 1750–1900, Scand. J. Hist., 39, 612–639, 2014.
Diaz, H. F., Kovats, R. S., McMichael, A. J., and Nicholls, N.: Climate and human health linkages on multiple timescales, in: History and Climate: Memories of the Future?, edited by Jones, P. D., Ogilvie, A. E. J., Davies, T. D., and Briffa, K. R., pp. 267–289, Springer, Berlin/Heidelberg, 2001.
Galloway, P. R.: Annual variations in deaths by age, deaths by cause, prices, and weather in London 1670 to 1830, Population Studies, 39, 487–505, 1985.
Galloway, P. R.: Long-term fluctuations in climate and population in the preindustrial era, Population and Development Review, 12, 1–24, 1986.
Galloway, P. R.: Secular changes in the short-term preventive, positive, and temperature checks to population growth in Europe, 1460 to 1909, Clim. Change, 26, 3–63, 1994.
Imhof, A. E.: Aspekte der Bevölkerungsentwicklung in den nordischen Ländern: 1720–1750, Francke, Giessen, 1976.
Lee, R. D.: Short-term variation: Vital rates, prices and weather , in: The Population History of England, edited by Wrigley, E. A. and Schofield, R. S., pp. 356–401, Cambridge University Press, Cambridge, 1981.
Ljungqvist, F. C., Seim, A., and Huhtamaa, H.: Climate and society in European history, Wiley Interdiscip. Rev. Clim. Change, 12, e691, 2021.
Mills, J. N., Gage, K. L., and Khan, A. S.: Potential influence of climate change on vector-borne and zoonotic diseases: a review and proposed research plan, Environ. Health Perspect., 118, 1507–1514, 2010.

Pfister, C. and Wanner, H.: Climate and Society in Europe: The Last Thousand Years, Bern: Haupt Verlag, 2021.

Vicedo-Cabrera, A. M., Scovronick, N., Sera, F., Royé, D., Schneider, R., Tobias, A., Astrom, C., Guo, Y., Honda, Y., Hondula, D., et al.: The burden of heat-related mortality attributable to recent human-induced climate change, Nature Climate Change, 11, 492–500, 2021.

Zhao, Q., Wen, B., Gasparrini, A., Tong, S., Overcenco, A., Urban, A., Schneider, A., Entezari, A., et al.: Global, regional, and national burden of mortality associated with short-term temperature variability from 2000 to 2019: a three-stage modelling study, The Lancet Planetary Health, 6, e410–e421, 2022.

---

## Author Comment (AC4)

**REVIEW 2**

The purpose of the paper by Liczbińska et al. was to examine the role of temperatures and precipitation on cause-specific mortality in nineteenth century Poznan. The authors' motivation was that a body of literature on the influence of temperature and precipitation in modern settings is emerging, and the authors want to examine whether temperature and precipitation also influenced mortality in past settings. The authors rightly state that the literature on climate and cause-specific mortality in historical Europe outside of infant mortality is limited (if not nearly absent), and their paper hence offers a novel contribution.

Thank you.

The authors' source material consists of parish registers for the city of Poznan between 1850 and 1900, including the date of death, place of residence and cause of death as variables. The causes of death are divided into three broad categories: airborne diseases, waterborne diseases, and other causes of death. They test the role of temperature and precipitation via a multinominal regression model, testing both for the influence of the same month and with a one-month lag. Finally, they also conduct a spatial analysis testing for the differences between neighbourhoods in Poznan. The authors conclude that the lagged monthly temperatures are a better predictor for both airborne and waterborne diseases, and that there were differences in mortality between the five quarters of Poznan during the study period.

The paper was an interesting read, but quite a few things confused me, and I have some major concerns about the authors' methodology and interpretation of results. The paper would furthermore immensely improve, if a native-English speaker would proofread it. I will address my comments and concerns below.

Thank you very much for the report. We have tried to accommodate Your remarks, recommendations, and advice as closely as we could.

**Major comments:**

1. The authors have divided the causes of death into three very broad categories called "airborne", "waterborne" and "other". I find this broad distinction problematic. From an epidemiological perspective there are monumental differences between respiratory epidemic diseases like measles and smallpox, endemic respiratory diseases like influenza, and complications like pneumonia. Moreover, some of the diseases in the "waterborne"-category are not exclusively waterborne. Typhoid fever can infect from person to person via close contact or via milk, tuberculosis (in the airborne-category) can also transmit via milk, dysentery is known to transmit via bad food products, and recent historical evidence from the 1853 cholera epidemic in Copenhagen suggests that cholera might also have infected from person to person in this setting (see: 1093/infdis/jix602 and https://doi.org/10.1371/journal.pntd.0006103).

   Moreover, the authors do not make explicit what the "others"-category contains. The paper would improve greatly, if the authors made explicit why they used these rough divisions, discuss the validity of using so broad categories, and considered the role of multiple routes of transmission.

2. We agree that the category "the other" should be specified ("other causes of deaths", with examples provided). This problem has already been emphasized by the first Reviewer.

   If infectious diseases were categorized in our work more precisely (e.g., respiratory epidemic diseases, endemic respiratory diseases; or measles, smallpox, etc.), our

goal would be **Out of the scope of statistical modeling** (the time series are not so long and there is not enough information to describe climatic effects for different diseases separately).
What we do in this work was to estimate the average effect for broader disease categories (in our work: waterborne diseases, airborne diseases) for which we can estimate definite temperature/ precipitation influence.

3.  In the methods-section, the authors state that they "explored various models" and found that the one-month lagged models yielded the best predictions in accordance with the AIC (l. 173-179). The way I read this, the authors' choice of one-month lags appears to be based on better fits to the data, and not based on biological or social explanations. The authors do not fully explain why the one-month lag is important, or why the non-lag is unimportant. I would strongly advise the authors to 1) include the results of the statistical analyses without lags too, and 2) expand on why the results of the non-lagged analyses are not important.
There must be some lag between the (high/low) temperature occurring in a given place and time and development of infectious agents in this place and time. Then there is a further delay between acquiring an infectious (or other) disease and developing of a clear clinical symptom allowing for diagnosis, then another delay between a clear clinical sign and death etc. We do not attempt to estimate the exact total delay (it is not possible with the time resolution of the data we have; it would not even be practical since we model broader disease/ death categories). Our goal is not to describe the exact disease dynamics (which would be better studied on current clinical data), but to test and estimate the magnitude and direction of climatic variables upon mortality due to waterborne and airborne diseases (AND to differentiate among the climatic effects upon these two categories). To this end, we do not want to dilute the temperature effect by not allowing for the time delay. Since the monthly temperature and one-month-lagged temperature (and similarly - precipitation) are correlated, we would find some (but weaker/diluted) effects even without realistic lagging.
So, the formalized model selection is necessary. We used AIC, as acknowledged in the paper. The winner is ABSOLUTELY clear (AIC=7899 for the one-month-lagged model we present and AIC=8479 for non-lagged model). Thank you for the suggestion, we will add the AIC comparison and discussion of the lag issue into the main body of the manuscript.

4.  The authors are right that overcrowding likely played a key role in the transmission of airborne diseases (l. 302-310). However, couldn't one also argue that crowding and general poverty played a key role for the so-called "waterborne" diseases? Many diarrhoeal diseases are known as "oral-fecal" diseases within epidemiology due to the importance of poor hygiene.
In the case of the poor epidemiological situation that took place in Poznań at that time, overcrowding (limited access to the already limited infrastructure, e.g., accesses to clean water, medical service, etc.) and poverty (worse standard of living, nutrition, lack of access to medical care, etc.) may additionally have contributed to the increase of mortality. The Market Square was the only place in the city where clean water through wooden pipes from the small Bogdanka River was supplied to public wells located there.
As we have emphasized in the text, the rest inhabitants drew water from shallow wells which were often polluted with harmful sewage discharged directly to the

rivers and the city moats. There were also primitive street gutters with ineffective drainage full of still, contaminated water. In many cases the Warta River was a source of drinking water.

5.  The paper starts with a climate-related angle, arguing that we need to examine whether temperatures and precipitation also influenced mortality patterns in past populations (l. 61-63). After that, the paper focuses exclusively on the historical demography and the context of nineteenth century Poland. As a historian, I do not find this problematic, but I would encourage the authors to either change the focus of the paper to be more climate-oriented, or to return to the modern relevance in the discussion and/or conclusion.
    Thank you for this suggestion. It has also been emphasized by the 1st Reviewer. We will propose a new title (**Climate and Disease in urban space (the evidence from 19th c. Poznań, Poland)** and the Introductory chapter will be re-edited.

6.  l. 280: "At the end of the 19th century there were almost 8,000 inhabitants per 1 km2 enclosed within the walls". Was the population density so high in all four quarters behind the city walls? If so, why are there so distinct differences between them? The authors mention that the poorest population lived in quarter 5, which was outside the city, whereas the city was populated by artisans. I am not an economic historian, but can it really be the case that the poorest exclusively lived outside the city walls? It seems a bit simplified to me.
    We cannot provide the exact population density in subsequent quarters of the city, because we do not have accurate data on their exact area. Historians have provided the area within the city walls which accounted of 943.4 ha. In (appx. 9.5 km2) and did not change till 1900. In the light of Prussian Statistics, in the 1860s, 1870s and 1890s the city was inhabited by over 53, 60 and 73 thousand people, respectively. It gives over 5,500, 6,300 and 7,700 people per km2, respectively. In 1900 the fortress was demolished, and suburban quarters were incorporated into the city. The urban area expanded to 3,300 ha and the population density decreased to 3,500 per km2.
    We have data on the number of inhabitants in each of 1-5 quarters, collected by the police heads during the cholera epidemic in 1866. Quarters 1, 2, 3 and 4 were inhabited by 8513, 8631, 11095 and 9194 people, respectively, plus troops (constituted of 12-15% of the total city's population). Meanwhile, Quarter 5 was inhabited by 7,706 people. The area outside the fortress did not have limited space, so theoretically it should not have been as crowded as the one within the fortress. In practice, people lived in one-room dwellings, often cramped, damp and unheated, one room being sometimes shared by 5 to 12 people. In 1866 Cholera Specific Mortality Rates in quarters 1, 2, 3, 4 and 5 were 20, 22, 28, 19 and 55 per 1,000 people, respectively. So, despite theoretically smaller population size in Q5, cholera death rates were the highest there. In quarter 5 people were deprived of medical care. The clinical symptoms of cholera were known to doctors working in 1866 since being described by Doctor Kaczkowski in 1830. Poznań inhabitants had already experienced cholera epidemics in 1831, 1837, 1848, 1852 and 1855, so that in 1866 it was not an unknown phenomenon for doctors and inhabitants of Q1-Q4 compared to migrants from Greater Poland villages, living in Q5. For the latter cholera could have been a new experience the did not know how to cope with.
    As for the social status of the inhabitants living in the city quarters, the professional status in the fortress and outside it was reconstructed earlier based on data from

parish books. Craftsmen and white-collar workers predominated in the fortress, while unskilled workers outside it. Unskilled workers earned the least. The workers earned an average of 500 to 600 marks a year, most of which they spent on rent: a man earned 1.6 marks a day, a woman 1.0 million marks, boys under the age of sixteen 0.75 marks, and girls - 0.50 marks (Łuczak 1965). While the salaries of skilled workers were sufficient to cover the cost of a very modest living, the earnings of unskilled workers did not provide them with a minimum subsistence. Even at the beginning of the 20th century, the rent for one room was 40 to 56% of the salary of the lowest earners. After paying it, a worker earning 300 mk a year had less than 200 mk left, so less than 1 mk per day was enough to support the whole family (Łuczak 1965). The intelligentsia (doctors, teachers, officials) lived within the walls of the fortress. The literature shows that the financial situation of this group was quite diverse: doctors and lawyers earned the best. There was a great polarization among officials: in 1847, the chief president of the office received 6,000 a year. thalers, while others - the so-called senior officials – from 800 to 1500 thalers. Rank-and-file officials, especially part-time employees, fared much worse (Makowski 1992). On the other hand, the group of good earners (about 1,000 tal./year) included middle school principals, teachers, and newspaper editors. The salaries of secondary school teachers ranged from 400 to 800 thalers per year.

7. My last major comment relates to the authors' explanations of the variability of the "waterborne" mortality between the quarters. The authors state that "Inhabitants of quarters: 1, 2 and 5 had access to the Warta River, being a potential source of drinking water for many of them. In those sectors the highest mortality due to waterborne diseases was observed" (l. 189-190). Figure 10 however shows that the probability of death was highest in quarters 1, 3 and 5 with a low probability of death in quarter 2. All three quarters had access to the river water, but with very different rates in mortality.

The surplus of deaths from waterborne diseases in quarter 1 (privileged in terms of the access to clean water) and quarter 3 (with no direct access to the rivers) was related to the presence of two hospitals in those quarters. Cholera patients were treated there and some of them died there, increasing the death statistics. We have mentioned this fact in the paper (lines 291-294).

**Minor comments:**
1. The authors lead the paper by stating that "Humans inhabiting a given climate zone for generations have developed particular characteristic traits that make them better suited to the environment. Due to genetic changes, individuals show adaptations in the structure and/or functioning of their organisms allowing them to live successfully in various environments" (l. 25-27). The authors are correct that some populations have adapted to specific diseases; people of African descent have better resistance to yellow fever and malaria, and Inuit populations of the Arctic regions are very vulnerable to influenza. However, genetics do not appear to be an issue in the paper, and it is not addressed further. If the authors believe that genetics was relevant for the paper, they should explicitly state so and include this in their discussion. Otherwise, it appears a bit redundant and could be omitted from the introduction.

We agree with the Reviewer's comment. This part is redundant and will be removed from the Introduction.

2. The authors are correct that the body of literature on the relationship between diseases, temperatures, precipitation and diseases in Europe is small (l. 79-80), it is not entirely absent. In recent years, a body of literature on the relationship between temperature, precipitation and malaria in Scandinavia has emerged. See: https://doi.org/10.1186/s12879-022-07422-2, doi: 1186/s12936-021-03744-9 and doi: 10.1186/1475-2875-8-94.
   Thank you for the hint. We will implement this paper into our work!

3. On the topic of infant mortality and temperatures (l. 64-65), the authors might want to reference the work by Johan Junkka in Sweden (doi: 1097/EE9.0000000000000176 and doi: 10.1016/j.envres.2020.110400).
   Thank you. The suggestion will be implemented.

4. On lines 77-78, the authors state that warm temperatures facilitate diarrhoeal diseases. This is true, and it is also worth noting that warm temperatures facilitated malaria (see above-cited malaria-papers) and plague (doi: 10.1098/rspb.2020.2725) in Europe historically.
   We agree with the Reviewer's comment. However, the indicated diseases did not occur in Poznań at all. This suggestion will be included in the introduction of the manuscript.

5. l. 94: change "till" to "until".
   Manuscript will be re-edited.

6. l. 100-101: "With time, the city started to suffer from the lack of free space and at the end of the 19th century it was virtually suffocating within the surrounding walls." Use a different phrasing than "virtually suffocating".
   Thank you for this suggestion. Manuscript will be re-edited.

7. l. 115-116: the authors need to revisit these two lines. First, they state that the urban ecology was "really bad". I would appreciate a different phrasing. Secondly, they state that "This translated into the health status of Poznań inhabitants, who often suffered from outbreaks of epidemics, i.e., cholera (Piankowski, 1988; Liczbińska, 2021)". The terms "epidemics" and "outbreaks" are synonymous for the same thing. It is also unclear to me, what the authors mean by "i.e., cholera". "i.e.," is an abbreviation for "id est", in English "that is".
   Yes, we agree with this hint. In stead of "This translated into the health status of Poznań inhabitants, who often suffered from outbreaks of epidemics, i.e., cholera (Piankowski, 1988; Liczbińska, 2021)" it should be "This translated into the health status of Poznań inhabitants, who often suffered from outbreaks of epidemics, e.x., cholera (Piankowski, 1988; Liczbińska, 2021)."

8. I don't presume that the authors meant that cholera was the only epidemic disease in Poznan, as they later mention scarlet fever, typhoid fever, and measles as other epidemic diseases.
   Cholera epidemics were not the only epidemics in Poznań but compared to the epidemics of smallpox or scarlet fever, for instance, they took the greatest mortality toll. They were the largest epidemics in the 19th-century Poznań causing massive and rapid changes in population numbers within a short period of time.

9. Finally, on line 116, the authors use the term "contagious diseases" rather than "infectious diseases", and "mortality toll" instead of "death toll".
Thank you for this suggestion. The suggestion will be implemented.

10. On lines 119-120, the authors state that "Infant mortality in Poznań was estimated as >250 deaths per 1,000 live births, and during the intervals of the epidemics, infant mortality was >300 per 1,000 live births". As I am sure the authors are aware, there was a massive drop in infant mortality rates during the second half of the nineteenth century. I would appreciate it, if the authors be precise and clarify which period they are referring to.
For example:
1850-1874 =315.03 per 1,000 live births*
1875-1884=280.4 per 1,000 live births**
1885-1894=261.8 per 1,000 live births**
1895-1904=199.9 per 1,000 live births**
1905-1913=171.05 per 1,000 live births**
*Calculated based of data derived from birth and death books for Poznań parishes
**Calculated based of data derived from Prussian Statistical Yearbooks

11. Lines 160-162: this sentence seems a bit redundant to me. The authors have already described the nineteenth century history of Poznan and can easily state that the city was divided into four quarters by the Prussian authorities. Also, it is unclear why the authors use the German name "Posen" in this sentence.
Posen was the official name of the city introduced by German authorities. This name was in force until 1918 (Poland gained independence from the partitions). Therefore, next to the name Poznań, Posen is often given in the brackets (German: Posen).

12. On line 165 the authors state that the five quarters of Poznan differed in ecological conditions. Please expand on this: how were they different?
Overall, the situation in Q1-5 was not black and white. In general, quarters 1-5 were inhabited mainly by craftsmen and white-collar workers, while quarter 5 – unskilled laborers. The ecological conditions in Poznań differed among the city quarters, which influenced the number of deaths from infectious diseases. The analyses of the epidemic in 1831 showed that on the right bank of Warta River (Q5) deaths from cholera accounted for 32.7% of all deaths in the city in 1831. In the light of the same data, deaths from cholera in the suburbs belonging to the parish of St. Martin (quarter 3) accounted for 25.2%, the parish of St. Mary Magdalene, encompassing the streets around the market square (quarter 1), accounted for 19.2% of all deaths. In 1866 Cholera Specific Mortality Rates in quarters 1, 2, 3, 4 and 5 were 20, 22, 28, 19 and 55 per 1,000 people, respectively. The poorest conditions were in the quarter 5, where people used shallow wells, often tainted with harmful sewage from cesspits, gutters, and rubbish sites. There were also primitive street gutters with ineffective drainage, which were full of stagnant contaminated water during summer heat periods. They did their laundry in the nearby rivers which was a source of drinking water. Inhabitants of quarters 2-4, although represented a wealthier part of Poznań society, had no access to clean water. Unfortunately, they used shallow wells located in courtyards. The further away from the market square, the worse was

the access to clean water intake. The quarter 1 had wells providing a supply of clean spring water but was not free of danger: the area around the Market Square was a place of a large rotation of people, and this was where travellers usually stayed. For example, cholera was probably brought to one of the taverns in the Old Market Square in June 1866 by rafters from Szczecin.
In the quarters 1 and 3 there were located hospitals. Their presence was helpful for the sick but on the other hand it could additionally have increased the number of deaths from cholera, or in general – the number of deaths from other causes in Q1 and Q3.

13. l. 229-231: "The incubation period of *Vibrio cholerae* is consistent with the duration of 1–5 days (Azman et al., 2013; Eisenberg et al., 2013). For diarrhoea-causing pathogens, mainly rotavirus and pathogenic *Escherichia coli*, the incubation period is usually around 1week (Eisenberg et al., 2003)…" I would suggest that the authors write "For **other** diarrhoea-causing pathogens", since cholera also produces diarrhoeal symptoms.
Thank you for this suggestion. The suggestion will be implemented.

14. l. 269-270: "This model suggests that there are substantial differences in the incidence of deaths due to waterborne and airborne diseases in different quarters." Incidence is an epidemiological term for the number of notified cases (not deaths) per capita. I would suggest that the authors go with "mortality rates" instead.
Thank you for this suggestion. The suggestion will be implemented.

15. l. 302: "The biological standard of living could…" You can just say "the standard of living".
This will be re-edited.

---

## Author Response (AR3)

October 4,  2023

Hans Linderholm
Editor
Climate of the Past

Dear Professor Linderholm,

Attached is a revision of our paper, *Climate and disease in historical urban space. Evidence from 19th-century Poznań, Poland*. Hopefully, we have satisfactorily addressed the concerns and recommendations of the Reviewers. Changes in the revision are indicated in BLUE font.

Responses to the comments of the reviewers are presented in the attachment to this letter (below).

Respectfully submitted,

Grażyna Liczbińska
and Co-authors

**REVIEW 1**

The article "Climate and Disease: the connection between temperature values and precipitation rates and the probability of death due to waterborne and airborne diseases in historical urban space (the evidence from Poznań, Poland)" by Liczbińska et al. is an important contribution to the growing number of European historical climate–health studies. It is especially welcome as most, to date, concerns regions of western Europe rather than central Europe. Indeed, a disproportionate number concerns conditions in the United Kingdom that, due to an early Industrialisation, is far from representative for 19th-century Europe as a whole.

Thank you very much for the report. We have tried to accommodate your remarks, recommendations, and advice as closely as we could.

**Major comments:**

The article focus too much, in its Introduction, on climatic health treats arising from high temperatures. There are numerous health threats, especially in the past, also arising from cold temperatures. The treat of cold temperatures should, in my opinion, be given an equal weight to treat of warm temperature in the background section. Especially in the historical climate–health studies literature, numerous examples of increased mortality following anomalously cold condition can be found that are presumably also highly relevant for Poland. Some suggestion of references to include are found below. I think the article needs to be better placed in a larger research context as many key works appears to be missing from the reference list.

Thank you very much for the review and for all your suggestions. The literature suggestions provided by the Reviewer have been implemented in the review **(page 1: lines 22–32 and page 2: lines 40–61).** We also take the comment as a motivation to think about possible future studies oriented to other extreme weather elements, including unusually low temperatures. This area is certainly interesting for future research but will need specialized data. For instance, in cities (like Poznan and unlike in smaller villages and solitary settlements), the effect of low temperatures was almost certainly filtered due to the city infrastructure for most but the lowest social strata. For instance, in Figures 4 and 5, we can see almost no effect of low temperatures on waterborne diseases (as expected) and only a relatively mild effect on airborne diseases.

In addition, I am a bit sceptical that that the living conditions in 19th century Poznań really were as poor as suggested by Liczbińska et al. This should be critically discussed as there is a huge disagreement among (economic) historians about the standards of living among the working poor in different parts of pre-industrial and early industrial Europe. Poland, admittedly, had relatively low standards of living compared to northwestern Europe, but I'm still not convinced that they actually were as poor as the article gives the impression of. To cite modern economic historical literature would be helpful to support the statements.

We realize that the phrase "bad conditions" can be very subjective. That is why we have provided a broader context on 19th century Poznań, which allows you to notice Poznan's delays in infrastructure compared to Western European cities. We have provided some examples as the moment of opening the water supply network in Poznań compared to Western Europe: from a 15-year delay compared to Berlin to over 50-year delay compared to Paris and London, or sewage systems: a delay of over 30 years (Prague) to over 50 years (Hamburg). Poznan's delays in civilizational achievements compared to other cities show that some aspects related to the

functioning of a large city had not been solved during the period studied **(page 4: lines 100–108 and p. 7: lines 224–231).**

**Minor comments:**
Title: The title is far too long. It needs to be shortened, preferably by half.
We agree with the Reviewer's remark. Our proposal for the new title is Climate and Disease in Urban Space. Evidence from 19th-century Poznań, Poland (**p. 1: lines 1–2**).

Line 13: This part of the sentence does not read well. Furthermore, "other diseases" is too vague.
We agree with the Reviewer's remark. We have replaced the third category "other diseases" with " other causes of death" and provided examples (**p. 5: lines 141 and 155; p. 6: lines 164–167**).

Lines 12, 16, 39 and other places: Better to write "temperature levels" and "precipitation amounts".
Thank you for your suggestions. They have been implemented (**p. 1: lines 10,  16–17**).

Line 16: In English, "block" rather than "quarter" are normally used in this context.
The suggestion has been implemented. The word "quarter" in this place has been replaced with "place of residence" (**p. 1: line 14**).

Lines 25–29: I would consider another introduction. The climate adaptation is, besides genetic (which takes a very long time), also social and cultural (and to some extent even socio-political). If the introduction is not changed this needs to be more clearly stated.
Thank you for this suggestion. The Introduction section has been totally rewritten **(pp. 1–2: lines 22–64).**

Line 53: Not only warm temperature. Weather conditions are also shaped by cold temperatures.
We agree with the Reviewer's suggestion. Information on the role of low temperatures has been implemented **(p. 1: lines 25–30; p. 2: lines 40–43 and 55–59).**

Lines 56–58: Not really relevant even to cite as the present article concerns Europe.
The Introduction has been rewritten and its parts have been removed **(pp. 1–2: lines 22–64).**

Line 62: Please consider using another word than "reasonably" here.
The sentence has been rewritten (**p. 2: lines 62–63).**

Lines 112–114: I would like to see more exact numbers here.
Explanation has been introduced to the main body of the manuscript (**p. 4: lines 100–107).**

Line 218: Different citation style than elsewhere.
The citation has been corrected **(currently p. 8: line 243).**

Line 223: I think "fungi" is too vague. There exists many different types of fungi.
Our intention was to list pathogens responsible for infectious diseases, i.e., infectious agents including microorganisms such as bacteria, viruses, fungi, and prions. We did not specify pathogenic fungi in this context.

Line 280: The number of inhabitants per square kilometre is not adding up with other information about the same thing in the article. Please, double-check this information.
This has been checked. In 1896, the area of the city enclosed within the walls was appx. 9.5 $km^2$. At that time the city had a population of 73,293 (After Galloway, Patrick R., 2007, "Galloway Prussia Database 1861 to 1914"). Our calculations demonstrate that there were over 7,700 people per 1 $km^2$ and are similar to the numbers provided by Statistics for the City of Poznań (ed. by Kruszka 2008; p. 53: 7,700-7,800 people per 1 $km^2$ ). Since the area of the city had not changed until 1900, in the 1860s there were more than 5,600 people per $km^2$, and in the 1870s - more than 6,400 people per $km^2$. After the fortress was demolished in 1900, suburban villages were incorporated into the city. The urban area expanded to 3,300 hectares and the population density decreased to 3,500 per $km^2$ **(p. 3: lines 93–99).**

Line 304 and other places: Please provide a distinction between "workers" and "labourers".
We openly acknowledge that we are not experts in this field. According to the Britannica Dictionary, the word "worker" refers to a person who does a specified type of work or who works in a specified way. The word "laborer" refers to a person who does unskilled physical work: a daily labourer; according to Longman Dictionary, someone whose work needs strength rather than skill. Changes have been made in the text: since quarter 5 was inhabited by unskilled groups, we have replaced the word "worker" with "laborer" **(p. 10: lines 307, 308).**

Lines 308–309: I am highly sceptical too that the conditions were so bad (except in exceptional circumstances). If this indeed is true – which I do not entirely rule out – it should be supported by more modern (post-Communistic period) scholarship.
We compared the conditions in the city of Poznań to other 19th-century European cities. Poznań was a city lagging behind in terms of modern infrastructure, such as waterworks and sewage systems. For example, the water supply network was opened in 1866 while in Berlin – 15 years earlier, in London and Paris - at the beginning of the 19th century. Sewage systems were opened in 1896, while in Hamburg in 1842. Delays in infrastructure development meant that living conditions were worse than in other Western European cities at the same time **(p. 7: lines 224–231).** Modern literature has been cited as well.

All the graphs: They needs to be improved and provided as proper vector graphics. The resolution is very poor. There size and dimensions also need to be streamlined. The font size in the graphs is, in general, too large.
Graphs have been changed in the review. **Their number has been reduced from 11 to 7.**

**Suggested references:**
Thank you. We have implemented the suggested literature items in the main body of the manuscript.

Åström, D. O., Edvinsson, S., Hondula, D., Rocklöv, J., and Schumann, B.: On the association between weather variability and total and cause-specific mortality before and during industrialization in Sweden, Demographic Research, 35, 991–1010, 2016.

Bradley, L.: An enquiry into seasonality in baptisms, marriages, and burials, Local Population Studies, 4, 21–40, 1970.

Calleja-Agius, J., England, K., and Calleja, N.: The effect of global warming on mortality, Early Human Development, 155, 105 222, 2021.

Castenbrandt, H.: A forgotten plague: dysentery in Sweden, 1750–1900, Scand. J. Hist., 39, 612–639, 2014.

Diaz, H. F., Kovats, R. S., McMichael, A. J., and Nicholls, N.: Climate and human health linkages on multiple timescales, in: History and Climate: Memories of the Future?, edited by Jones, P. D., Ogilvie, A. E. J., Davies, T. D., and Briffa, K. R., pp. 267–289, Springer, Berlin/Heidelberg, 2001.

Galloway, P. R.: Annual variations in deaths by age, deaths by cause, prices, and weather in London 1670 to 1830, Population Studies, 39, 487–505, 1985.

Galloway, P. R.: Long-term fluctuations in climate and population in the preindustrial era, Population and Development Review, 12, 1–24, 1986.

Galloway, P. R.: Secular changes in the short-term preventive, positive, and temperature checks to population growth in Europe, 1460 to 1909, Clim. Change, 26, 3–63, 1994.

Imhof, A. E.: Aspekte der Bevölkerungsentwicklung in den nordischen Ländern: 1720–1750, Francke, Giessen, 1976.

Lee, R. D.: Short-term variation: Vital rates, prices and weather , in: The Population History of England, edited by Wrigley, E. A. and Schofield, R. S., pp. 356–401, Cambridge University Press, Cambridge, 1981.

Ljungqvist, F. C., Seim, A., and Huhtamaa, H.: Climate and society in European history, Wiley Interdiscip. Rev. Clim. Change, 12, e691, 2021.

Mills, J. N., Gage, K. L., and Khan, A. S.: Potential influence of climate change on vector-borne and zoonotic diseases: a review and proposed research plan, Environ. Health Perspect., 118, 1507–1514, 2010.

Pfister, C. and Wanner, H.: Climate and Society in Europe: The Last Thousand Years, Bern: Haupt Verlag, 2021.

Vicedo-Cabrera, A. M., Scovronick, N., Sera, F., Royé, D., Schneider, R., Tobias, A., Astrom, C., Guo, Y., Honda, Y., Hondula, D., et al.: The burden of heat-related mortality attributable to recent human-induced climate change, Nature Climate Change, 11, 492–500, 2021.

Zhao, Q., Wen, B., Gasparrini, A., Tong, S., Overcenco, A., Urban, A., Schneider, A., Entezari, A., et al.: Global, regional, and national burden of mortality associated with short-term temperature variability from 2000 to 2019: a three-stage modelling study, The Lancet Planetary Health, 6, e410–e421, 2022.

**REVIEW 2**

The purpose of the paper by Liczbińska et al. was to examine the role of temperatures and precipitation on cause-specific mortality in nineteenth century Poznan. The authors' motivation was that a body of literature on the influence of temperature and precipitation in modern settings is emerging, and the authors want to examine whether temperature and precipitation also influenced mortality in past settings. The authors rightly state that the literature on climate and cause-specific mortality in historical Europe outside of infant mortality is limited (if not nearly absent), and their paper hence offers a novel contribution.

Thank you.

The authors' source material consists of parish registers for the city of Poznan between 1850 and 1900, including the date of death, place of residence and cause of death as variables. The causes of death are divided into three broad categories: airborne diseases, waterborne diseases, and other causes of death. They test the role of temperature and precipitation via a multinominal regression model, testing both for the influence of the same month and with a one-month lag. Finally, they also conduct a spatial analysis testing for the differences between neighbourhoods in Poznan. The authors conclude that the lagged monthly temperatures are a better predictor for both airborne and waterborne diseases, and that there were differences in mortality between the five quarters of Poznan during the study period.

The paper was an interesting read, but quite a few things confused me, and I have some major concerns about the authors' methodology and interpretation of results. The paper would furthermore immensely improve, if a native-English speaker would proofread it. I will address my comments and concerns below.

Thank you very much for the report. We have tried to accommodate your remarks, recommendations, and advice as closely as we could. The English version has been proofread by a native-English speaker.

**Major comments:**

1. The authors have divided the causes of death into three very broad categories called "airborne", "waterborne" and "other". I find this broad distinction problematic. From an epidemiological perspective there are monumental differences between respiratory epidemic diseases like measles and smallpox, endemic respiratory diseases like influenza, and complications like pneumonia. Moreover, some of the diseases in the "waterborne"-category are not exclusively waterborne. Typhoid fever can infect from person to person via close contact or via milk, tuberculosis (in the airborne-category) can also transmit via milk, dysentery is known to transmit via bad food products, and recent historical evidence from the 1853 cholera epidemic in Copenhagen suggests that cholera might also have infected from person to person in this setting (see: 1093/infdis/jix602 and https://doi.org/10.1371/journal.pntd.0006103).

   Moreover, the authors do not make explicit what the "others"-category contains. The paper would improve greatly, if the authors made explicit why they used these rough divisions, discuss the validity of using so broad categories, and considered the role of multiple routes of transmission.

   If in our work infectious diseases were categorized more precisely (e.g., respiratory epidemic diseases, endemic respiratory diseases; or measles, smallpox, etc.), our goal would be **out of the scope of statistical modeling** (the time series are not

so long and there is not enough information to describe climatic effects for different diseases separately).
**What we did in this work** was to estimate the average effect for broader disease categories (in our work: waterborne diseases, airborne diseases) for which we can estimate definite temperature/ precipitation influence.
We agree that the category "the other" should be specified ("other causes of deaths", with examples provided) **(p. 5: lines 141–142).**

2. In the methods-section, the authors state that they "explored various models" and found that the one-month lagged models yielded the best predictions in accordance with the AIC (l. 173-179). The way I read this, the authors' choice of one-month lags appears to be based on better fits to the data, and not based on biological or social explanations. The authors do not fully explain why the one-month lag is important, or why the non-lag is unimportant. I would strongly advise the authors to 1) include the results of the statistical analyses without lags too, and 2) expand on why the results of the non-lagged analyses are not important.
It is even a priori clear that there must be some lag between (high/low) temperature occurring in a given place and time and the development of infectious agents in this place and time. Then there is a further delay between acquiring an infectious (or other) disease and developing clear clinical symptoms allowing for diagnosis, then another delay between a clear clinical sign and death, etc. **We do not attempt to estimate the exact total delay** (it is not possible with the time resolution of the data we have; it would not even be practical since we modeled broader diseases/ causes of death categories). Our goal is not to describe **the exact disease dynamics** (which would be better studied on current clinical data) **but to test and estimate the magnitude and direction of climatic variables' effects upon mortality due to waterborne and airborne diseases** (and to differentiate among the climatic effects upon these two categories). To this end, we do not want to dilute the temperature effect by not allowing for the time delay. Since the monthly temperature and one-month lag (and similarly - precipitation) correlated, we would find some (but weaker/diluted) effects even without realistic lagging.
So, the formalized model selection is necessary. We used AIC, as acknowledged in the paper: AIC=7899 for the one-month-lagged model we present and AIC=8479 for a non-lagged model. Thank you for the suggestion, we have added the AIC comparison to the main body of the manuscript **(p. 6: lines 181–182).**

3. The authors are right that overcrowding likely played a key role in the transmission of airborne diseases (l. 302-310). However, couldn't one also argue that crowding and general poverty played a key role for the so-called "waterborne" diseases? Many diarrhoeal diseases are known as "oral-fecal" diseases within epidemiology due to the importance of poor hygiene.
In the case of the poor epidemiological situation that took place in Poznań during the period studied, overcrowding (limited access to the already limited infrastructure, e.g., access to clean water, medical service, etc.) and poverty (worse standard of living, poor nutrition, lack of access to medical care, etc.) might have additionally contributed to the increase of mortality. Poverty was linked to malnutrition, and this affected the reduction of immunity, poverty significantly limited access to medical care and medicines, while poor living conditions prolonged recovery, posed the risk of illness, deterioration of health (damp, crowded rooms), etc. Poverty might have favored morbidity and mortality due to infectious diseases

in general – including waterborne diseases. In Poznań, the poorest districts were completely deprived of infrastructure because the city did not invest in these places. This resulted in a lack of hygiene. Cholera mortality rates (added to the article) in the poorest district were more than twice as high as in other parts of Poznań.

The paper starts with a climate-related angle, arguing that we need to examine whether temperatures and precipitation also influenced mortality patterns in past populations (l. 61-63). After that, the paper focuses exclusively on the historical demography and the context of nineteenth century Poland. As a historian, I do not find this problematic, but I would encourage the authors to either change the focus of the paper to be more climate-oriented, or to return to the modern relevance in the discussion and/or conclusion.

Thank you for this suggestion. Some parts of the paper have been rewritten. Unfortunately, we do not have any works about the climate in 19th-century Poland. This work is the first attempt to combine historical demography with epidemiology and climate. The specificity of Poznań (enclosed within a fortress), delays in infrastructural development and an increase in the number of inhabitants, vicinity to the rivers had an important impact on the course of epidemics which hit the city, and climatic conditions might have additionally favored them, as we have shown in the case of waterborne diseases.

4. l. 280: "At the end of the 19th century there were almost 8,000 inhabitants per 1 km2 enclosed within the walls". Was the population density so high in all four quarters behind the city walls? If so, why are there so distinct differences between them? The authors mention that the poorest population lived in quarter 5, which was outside the city, whereas the city was populated by artisans. I am not an economic historian, but can it really be the case that the poorest exclusively lived outside the city walls? It seems a bit simplified to me.

We cannot provide the exact population density in subsequent quarters of the city because we do not have accurate data on population size in each area. Historians have provided that the area within the city walls accounted for 9.43 km$^2$ and did not change until 1900. In the light of Prussian Statistics, in the 1860s, 1870s, and 1890s the city was inhabited by over 53, 60, and 73 thousand people, respectively, which gives over 5,500, 6,300, and 7,700 people per km$^2$, respectively. In 1900 the fortress was demolished, and suburban areas were incorporated into the city. The urban area expanded to 3,300 ha and the population density decreased to 3,500 per km$^2$. We do not have data on the number of inhabitants in quarters 1-5, except in the year 1866. During the cholera epidemic in 1866, such data were collected by the police heads. Quarters 1, 2, 3, and 4 were inhabited by 8513, 8631, 11095, and 9194 people, respectively, while quarter 5 - by 7,706 people. The values of Cholera Specific Mortality Rates in 1866 were in quarters 1, 2, 3, and 4 around 2 deaths per 100 people, while in quarter 5 – there were 6 deaths per 100 people. So, in quarter 5, despite the smaller population size, the values were as twice high as in Q1–Q4. The clinical symptoms of cholera were known to doctors working in 1866 since they had been described by Doctor Kaczkowski in 1830. Poznań inhabitants had already experienced cholera epidemics in 1831, 1837, 1848, 1852, and 1855, so in 1866 it was not an unknown phenomenon for doctors and inhabitants of Q1-Q4 compared to migrants from Greater Poland villages, living in Q5. For the latter cholera could have been a new experience, and they did not know how to cope with it. The same might have been done with other diseases.

The spatial distribution of Poznan inhabitants presented in this paper is to some extent a simplification. In the light of the work of historians, the left bank of the river was inhabited by a wealthier part of Poznan society (officials, merchants, craftsmen from generation to generation), while the right bank – by the lowest strata, mainly migrants from sub-Poznań villages, without any skilled profession. Since the area on the left bank of the Warta River was unable to absorb outsiders/ migrants, they settled on the right bank. The social status of the inhabitants living in the city quarters was reconstructed in earlier work (Liczbińska 2009b). Craftsmen and white-collar workers predominated in quarters 1–4, while laborers – in 5. Unskilled laborers earned the least: 500 to 600 marks a year on average, and most of their salaries were spent on rent. A man earned 1.6 marks a day, a woman 1 mark, boys below the age of 16 years – 0.75 marks, and girls – 0.50 marks. The earnings of laborers did not provide them with a minimum subsistence. Even at the beginning of the 20th century, the rent for one room was 40 to 56% of the salary of the lowest earners. The intelligentsia (doctors, teachers, officials) and craftsmen lived on the left bank of the Warta River.

5. My last major comment relates to the authors' explanations of the variability of the "waterborne" mortality between the quarters. The authors state that "Inhabitants of quarters: 1, 2 and 5 had access to the Warta River, being a potential source of drinking water for many of them. In those sectors the highest mortality due to waterborne diseases was observed" (l. 189-190). Figure 10 however shows that the probability of death was highest in quarters 1, 3 and 5 with a low probability of death in quarter 2. All three quarters had access to the river water, but with very different rates in mortality.
The surplus of deaths from waterborne diseases in quarter 1 and quarter 3 might have been related to the presence of two hospitals in those quarters. Cholera patients were treated there and some of them died in hospitals, increasing the death statistics. We have mentioned this fact in the paper **(p. 9: lines 282–293).**

**Minor comments:**
1. The authors lead the paper by stating that "Humans inhabiting a given climate zone for generations have developed particular characteristic traits that make them better suited to the environment. Due to genetic changes, individuals show adaptations in the structure and/or functioning of their organisms allowing them to live successfully in various environments" (l. 25-27). The authors are correct that some populations have adapted to specific diseases; people of African descent have better resistance to yellow fever and malaria, and Inuit populations of the Arctic regions are very vulnerable to influenza. However, genetics do not appear to be an issue in the paper, and it is not addressed further. If the authors believe that genetics was relevant for the paper, they should explicitly state so and include this in their discussion. Otherwise, it appears a bit redundant and could be omitted from the introduction.
We agree with the Reviewer's comment. This part was redundant and has been removed from the Introduction. Some parts of the Introduction have been rewritten.

2. The authors are correct that the body of literature on the relationship between diseases, temperatures, precipitation and diseases in Europe is small (l. 79-80), it is not entirely absent. In recent years, a body of literature on the relationship

between temperature, precipitation and malaria in Scandinavia has emerged. See: https://doi.org/10.1186/s12879-022-07422-2, doi: 1186/s12936-021-03744-9 and doi: 10.1186/1475-2875-8-94.

Thank you for this hint. These papers have been implemented into our work **(p. 2: lines 53–55).** In the city of Poznań in the period under study, there were recorded 2 cases of death due to malaria: a 15-year-old boy died in 1855, and a 7-month-old girl died in 1890.

3. On the topic of infant mortality and temperatures (l. 64-65), the authors might want to reference the work by Johan Junkka in Sweden (doi: 1097/EE9.0000000000000176 and doi: 10.1016/j.envres.2020.110400).

Thank you. The suggestions/ literature have been implemented **(p. 2: lines 59–60).**

4. On lines 77-78, the authors state that warm temperatures facilitate diarrhoeal diseases. This is true, and it is also worth noting that warm temperatures facilitated malaria (see above-cited malaria-papers) and plague (doi: 10.1098/rspb.2020.2725) in Europe historically.

We agree with the Reviewer's comment. We have included the proposed literature **(p. 2: lines 53–55).** However, in Poznań in the period under study, there were 2 cases of malaria recorded in death books.

5. l. 94: change "till" to "until".

The sentence has been re-edited.

6. l. 100-101: "With time, the city started to suffer from the lack of free space and at the end of the 19th century it was virtually suffocating within the surrounding walls." Use a different phrasing than "virtually suffocating".

Thank you for this suggestion. The sentence has been re-edited.

7. l. 115-116: the authors need to revisit these two lines. First, they state that the urban ecology was "really bad". I would appreciate a different phrasing. Secondly, they state that "This translated into the health status of Poznań inhabitants, who often suffered from outbreaks of epidemics, i.e., cholera (Piankowski, 1988; Liczbińska, 2021)". The terms "epidemics" and "outbreaks" are synonymous for the same thing. It is also unclear to me, what the authors mean by "i.e., cholera". "i.e.," is an abbreviation for "id est", in English "that is".

Yes, we agree with this hint. This part of the text has been rewritten **(p. 4: lines 100–105).**

8. I don't presume that the authors meant that cholera was the only epidemic disease in Poznan, as they later mention scarlet fever, typhoid fever, and measles as other epidemic diseases.

Cholera epidemics were not the only epidemics in Poznań but compared to the epidemics of smallpox or scarlet fever, for instance, they took the greatest mortality toll. They were the largest epidemics in the 19th-century Poznań causing massive and rapid changes in population numbers within a short period of time. This information has been implemented in the chapter: *The city of Poznań in the 19th century* **(p. 4: lines 100–105).**

9. Finally, on line 116, the authors use the term "contagious diseases" rather than "infectious diseases", and "mortality toll" instead of "death toll".
Thank you for this suggestion. The suggestion has been implemented.

10. On lines 119-120, the authors state that "Infant mortality in Poznań was estimated as >250 deaths per 1,000 live births, and during the intervals of the epidemics, infant mortality was >300 per 1,000 live births". As I am sure the authors are aware, there was a massive drop in infant mortality rates during the second half of the nineteenth century. I would appreciate it, if the authors be precise and clarify which period they are referring to.

11. We have calculated the values of infant mortality rates and their changes are presented below. Some figures have been implemented into the paper **(p. 4: lines 105–108).**

    1850-1874 =315.03 per 1,000 live births*
    1875-1884=280.4 per 1,000 live births**
    1885-1894=261.8 per 1,000 live births**
    1895-1904=199.9 per 1,000 live births**
    1905-1913=171.05 per 1,000 live births**
    *Calculated based on data derived from birth and death books for Poznań parishes
    **Calculated based on data derived from Prussian Statistical Yearbooks
    The values have been implemented into the main body of the text.

12. Lines 160-162: this sentence seems a bit redundant to me. The authors have already described the nineteenth century history of Poznan and can easily state that the city was divided into four quarters by the Prussian authorities. Also, it is unclear why the authors use the German name "Posen" in this sentence.
Posen was the official name of the city introduced by German authorities. This name was in force until 1918 (Poland gained independence from the partitions). Therefore, next to the name Poznań, Posen is often given in the brackets (German: Posen; see now). This unnecessary name was excluded from the text.

13. On line 165 the authors state that the five quarters of Poznan differed in ecological conditions. Please expand on this: how were they different?
Overall, the situation in quarters 1-5 was not black and white. In general, quarters 1-4 were inhabited mainly by craftsmen and white-collar workers, while quarter 5 – unskilled laborers. The ecological conditions in Poznań differed among the city quarters, which influenced the number of deaths from infectious diseases. The analyses of the cholera epidemic in 1831 showed that on the right bank of Warta River (quarter 5) deaths from cholera accounted for 32.7% of all deaths in the city in this year. Meanwhile, deaths from cholera in the parish of St. Martin (quarter 3) accounted for 25.2%, while the parish of St. Mary Magdalene, encompassing the streets around the market square (quarter 1), accounted for 19.2% of all deaths. In 1866 Cholera Specific Mortality Rates in quarters 1, 2, 3, 4, and 5 were 2, 2, 3, 3, and 6 per 100 people, respectively. The poorest conditions were in quarter 5, where people used shallow wells, often tainted with harmful sewage from cesspits, gutters, and rubbish sites. There were also primitive street gutters with ineffective drainage, which were

full of stagnant contaminated water during summer heat periods. They did their laundry in the nearby rivers which were a source of drinking water.

Although Inhabitants of quarters 2-4 did not live in the vicinity of the rivers and represented a wealthier part of Poznań society, they did not have access to clean water. They used shallow wells located in courtyards.

In the quarters 1 and 3, there were located hospitals. Their presence was helpful for the sick but on the other hand, it could additionally have increased the number of deaths from cholera, or in general – the number of deaths from other causes in Q1 and Q3. This information has been added to the main text.

14. l. 229-231: "The incubation period of *Vibrio cholerae* is consistent with the duration of 1–5 days (Azman et al., 2013; Eisenberg et al., 2013). For diarrhoea-causing pathogens, mainly rotavirus and pathogenic *Escherichia coli*, the incubation period is usually around 1week (Eisenberg et al., 2003)…" I would suggest that the authors write "For **other** diarrhoea-causing pathogens", since cholera also produces diarrhoeal symptoms.

Thank you for this suggestion. The suggestion has been implemented **(p. 7, line 204).**

15. l. 269-270: "This model suggests that there are substantial differences in the incidence of deaths due to waterborne and airborne diseases in different quarters." Incidence is an epidemiological term for the number of notified cases (not deaths) per capita. I would suggest that the authors go with "mortality rates" instead.

Thank you for this suggestion. The suggestion has been implemented **(p. 9: line 264 and 296).**

16. l. 302: "The biological standard of living could…" You can just say "the standard of living".

This has been rewritten **(p. 10: line 303).**

November 19,  2023

Hans Linderholm

Editor

Climate of the Past

Dear Professor Linderholm**,**

Attached is a revision of our paper, *Climate and disease in historical urban space. Evidence from 19th-century Poznań, Poland*. Hopefully, we have satisfactorily addressed the concerns and recommendations of the Second Reviewer. Changes in the revision are indicated in RED font.

Responses to the comments of the reviewer are presented in the attachment to this letter (below).

Respectfully submitted,

Grażyna Liczbińska
and Co-authors

**Report #2**

I have two minor suggestions for improvements:

- Section 3 on the background of meteorological observations in Poznan is very long with detailed information on the changing street locations of the weather stations used to measure temperature and precipitation. If this has relevance for the quality of the measurements, it should be explicitly stated in the section. Right now, it is unclear to me why we need to know in detail where the weather station was located.

This has been done. See page 4, lines 110–125.

- Section 2, footnotes 1-3: The authors have estimated infant mortality rates based on parish registers and Prussian statistical yearbooks. I wonder whether it would be possible for the authors to include the results of those calculations as supplementary material to the article.

This has been done. See page 4, lines 106–108. Tables S1–S2 were added as Supplementary materials.